# A General Representation-Based Approach to Multi-Source Domain Adaptation

Ignavier Ng [* 1]   Yan Li [* 2]   Zijian Li [1 2]   Yujia Zheng [1]   Guangyi Chen [1 2]   Kun Zhang [1 2]

## Abstract

A central problem in unsupervised domain adaptation is determining what to transfer from labeled source domains to an unlabeled target domain. To handle high-dimensional observations (e.g., images), a line of approaches use deep learning to learn latent representations of the observations, which facilitate knowledge transfer in the latent space. However, existing approaches often rely on restrictive assumptions to establish identifiability of the joint distribution in the target domain, such as independent latent variables or invariant label distributions, limiting their real-world applicability. In this work, we propose a general domain adaptation framework that learns compact latent representations to capture distribution shifts relative to the prediction task and address the fundamental question of what representations should be learned and transferred. Notably, we first demonstrate that learning representations based on all the predictive information, i.e., the label's Markov blanket in terms of the learned representations, is often underspecified in general settings. Instead, we show that, interestingly, general domain adaptation can be achieved by partitioning the representations of Markov blanket into those of the label's parents, children, and spouses. Moreover, its identifiability guarantee can be established. Building on these theoretical insights, we develop a practical, nonparametric approach for domain adaptation in a general setting, which can handle different types of distribution shifts.

## 1. Introduction

Unsupervised domain adaptation aims to transfer knowledge from labeled source domains to an unlabeled target domain, particularly in scenarios where the training and testing data distributions differ substantially. In a multi-source domain adaptation (MSDA) setup, each source domain $u \in \{1, \ldots, M\}$ provides access to a labeled dataset $(\mathbf{x}^{(u)}, \mathbf{y}^{(u)}) = \{(\mathbf{x}_k^{(u)}, y_k^{(u)})\}_{k=1}^{m_u}$, where $m_u$ represents the number of samples in domain $u$. Here, the $i$-th dimension of the feature vector $X$ is denoted as $X_i$, and $x_{ik}^{(u)}$ corresponds to the value of the $i$-th feature for the $k$-th sample in domain $u$. The goal is to train a classifier that generalizes to an unlabeled target domain, where only the feature vectors $\mathbf{x}^{\tau} = \{\mathbf{x}_k^{\tau}\}_{k=1}^{m}$ are available.

Determining the joint distribution $P_{X,Y}^{\tau}$ in the target domain based solely on the marginal distribution $P_X^{\tau}$ is a fundamentally underdetermined problem. In the absence of additional assumptions, there are infinitely many possible joint distributions $P_{X,Y}^{\tau}$ that can align with the observed marginal distribution. Therefore, assumptions that connect the source and target domain distributions are essential for identifying the target joint distribution. Common approaches impose constraints to ensure a degree of similarity across these distributions. A widely adopted assumption is covariate shift (Pan & Yang, 2009), which asserts that the conditional distribution $P_{Y|X}$ remains consistent across domains while the marginal feature distribution $P_X$ varies. Alternatively, other frameworks account for variations in $P_Y$ or assume that transformations between the source and target features are linear (Zhang et al., 2015), offering additional ways to model domain relationships.

To avoid restrictive parametric assumptions about the relationships between domains, the principle of minimal changes is often considered (Schölkopf et al., 2012; Zhang et al., 2013). This perspective is particularly effective when analyzed through the lens of the data generating process. For instance, when the underlying process is $Y \rightarrow X$, the conditional distributions $P_Y$ and $P_{X|Y}$ can vary independently across domains. By factoring the joint distribution in this way, domain shifts can be represented in a parsimonious and structured manner. Moreover, changes in $P_{X|Y}$ are often constrained to lie on a low-dimensional manifold, further simplifying the problem (Stojanov et al., 2019). Advances in domain adaptation frameworks, particularly those leveraging multiple-domain data, have demonstrated the feasibility of uncovering the data-generating process and capturing these domain shifts (Huang et al., 2020; Zhang et al., 2020).

---

[*]Equal contribution [1]Carnegie Mellon University [2]Mohamed bin Zayed University of Artificial Intelligence.

*Proceedings of the 42$^{nd}$ International Conference on Machine Learning*, Vancouver, Canada. PMLR 267, 2025. Copyright 2025 by the author(s).

With the increasing capabilities of deep learning, another prominent line of work leverages neural architectures to map high-dimensional features into a latent representation space, ensuring that the latent variables $Z$ are marginally invariant across domains. This approach is motivated by efficiency: by working in a lower-dimensional latent space, it aligns with the principle of minimal changes, as it only models the essential domain shifts while discarding irrelevant variations. A classifier can then be trained on the labeled source data to ensure that the latent space retains predictive information about the labels (Ben-David et al., 2010; Ganin & Lempitsky, 2015; Zhao et al., 2018; Li et al., 2024a). While this strategy enables domain alignment in the latent space, the joint distributions $P_{Z,Y}$ may still vary significantly across domains, potentially degrading performance in the target domain. To address this, several works employ generative models or disentanglement techniques for the latent representations (Cai et al., 2019a; Lu et al., 2021; Yin et al., 2025). However, these methods typically rely on assumptions on the data distributions such as exponential family, or lack guarantees of identifiability for the target joint distribution $P_{X,Y}^\tau$ or the learned representations, limiting their ability to recover the true data generating process. The lack of identifiability raises concerns about the trustworthiness and reliability of these approaches, particularly when applied to real-world scenarios involving complex domain shifts.

Recent works by Kong et al. (2022) and Li et al. (2024b) have introduced theoretical frameworks that establish different types of identifiability results for latent representations in domain adaptation. They partition the latent space into different subspaces according to its connection with domains or labels. Although different types of identifiability results have been provided for identifying the latent representations and joint distribution in the target domain, these works often rely on restrictive assumptions such as independent latent variables or invariant label distributions, limiting their real-world applicability.

In this work, we propose a general domain adaptation framework that learns compact latent representations to capture distribution shifts relative to the prediction task and address the fundamental question of what representations should be learned and transferred. Notably, we first demonstrate that learning representations based on all the predictive information, such as the label's Markov blanket in terms of the learned representations, is often underspecified for domain adaptation in general settings. Instead, we show that, interestingly, general domain adaptation can be achieved by partitioning the representations of Markov blanket into those of the label's parents, children, and spouses. Accordingly, we establish identifiability of the joint distribution in the target domain, by learning low-dimensional representations of the changing distributions. Building on these theoretical insights, we develop a practical, nonparametric framework

for domain adaptation in a general setting, which can handle different types of distribution shifts. Finally, we validate our framework on real-world datasets, demonstrating that it outperforms existing methods.

## 2. Related Works

### 2.1. Domain Adaptation

Domain adaptation (Patel et al., 2015; Wilson & Cook, 2020; Farahani et al., 2021) aims to transfer knowledge from labeled source domains to an unlabeled target domain, such that the model can generalize to the target domain. A classical approach is to learn domain-invariant representations (Ganin & Lempitsky, 2015; Bousmalis et al., 2016), which are extracted by aligning the features across different domains. For instance, Long et al. (2017; 2018) applied maximum mean pseudo-labels and kernel methods for domain alignment, while Tzeng et al. (2014) adopt an adaptation layer and domain confusion loss to learn domain-invariant representations.

A different line of works rely on the assumption that conditional distributions $P(Z \mid Y)$ remain stable across domains, enabling the extraction of domain-invariant representations for each class (Chen et al., 2019b;a; Kang et al., 2020). For instance, Xie et al. (2018) minimize inter-class domain discrepancy, while Shu et al. (2018) constrains boundaries to avoid high-density regions via virtual adversarial domain adaptation. Target shift, where $P_Y$ varies across domains, has also been widely studied (Zhang et al., 2013; Lipton et al., 2018; Wen et al., 2020; Garg et al., 2020; Roberts et al., 2022). For instance, Tachet des Combes et al. (2020) developed theoretical guarantees for the transfer performance under generalized label shift, while Shui et al. (2021) proposed selecting relevant source domains based on conditional distribution similarity.

Recent works incorporate causality into domain adaptation (Kong et al., 2022; Magliacane et al., 2018; Teshima et al., 2020; Chen & Bühlmann, 2021; Gong et al., 2016; Stojanov et al., 2019). For instance, Zhang et al. (2013; 2015) investigated target shift, conditional shift, and generalized target shift by assuming independent change for $P(Y)$ and $P(X \mid Y)$. Cai et al. (2019a) learned disentangled semantic representations by leveraging causal generation process, while Stojanov et al. (2021) showed that domain-invariant features require domain knowledge, giving rise to their proposed domain-specific adversarial networks. These methods typically require restrictive assumptions and are not able to identify the latent variables with theoretical guarantees.

### 2.2. Identification of Latent Variables

The identifiability of latent variables remains a fundamental challenge, as they are generally unidentifiable without addi-

tional assumptions (Hyvärinen & Pajunen, 1999; Locatello et al., 2019). In the case of a linear mapping from latent to observed variables—known as independent component analysis (ICA)—identifiability can be achieved by assuming non-Gaussian latent variables (Comon, 1994; Hyvarinen et al., 2002). However, relaxing the linearity assumption leads to the ill-posed problem of nonlinear ICA (Hyvärinen & Pajunen, 1999; Hyvärinen et al., 2023).

To address this, existing nonlinear ICA methods typically rely on sufficient variations in the latent variable distribution, often introduced through auxiliary variables such as time or domain indices (Hyvarinen & Morioka, 2016; 2017; Hyvarinen et al., 2019; Khemakhem et al., 2020). Alternative approaches constrain the mixing function, either by restricting it to specific function classes (Hyvärinen & Pajunen, 1999; Taleb & Jutten, 1999; Gresele et al., 2021; Buchholz et al., 2022) or enforcing sparsity (Zheng et al., 2022).

More recently, causal representation learning has extended beyond ICA by considering causally-related latent variables instead of independent ones (Schölkopf et al., 2021). Similar to nonlinear ICA, many approaches in this area leverage sufficient variations in the latent variable distributions, typically induced by interventions (Ahuja et al., 2023; Squires et al., 2023; von Kügelgen et al., 2023; Jiang & Aragam, 2023; Zhang et al., 2023; Varici et al., 2023; Varıcı et al., 2024a;b; Jin & Syrgkanis, 2023; Bing et al., 2024; Zhang et al., 2024), temporal data (Yao et al., 2022a;b; Lippe et al., 2022; 2023), or both (Lachapelle et al., 2022; 2024). Other approaches rely on counterfactual view (Brehmer et al., 2022), multi-view data (Yao et al., 2024; Xu et al., 2024), more supervision information (Yang et al., 2021; Shen et al., 2022; Liang et al., 2023), causal ordering prior (Kori et al., 2023), constraint on the latent support (Ahuja et al., 2023; Wang & Jordan, 2021), or structural constraints (Silva et al., 2006; Xie et al., 2020; Cai et al., 2019b; Xie et al., 2022; Adams et al., 2021; Huang et al., 2022; Dong et al., 2023; Kivva et al., 2021).

## 3. A Generative Model with Distribution Shift

We assume that the $d$-dimensional feature vector $X$ (e.g., image pixels) is generated from latent variables $Z = (Z_1, \ldots, Z_n)$ via an unknown, smooth, and invertible mixing function $g : \mathbb{R}^n \to \mathbb{R}^d$. Also, the label $Y$ is a categorical value that takes values from $v_1, \ldots, v_C$. In each domain, the latent variables $Z$ and the label $Y$ are governed by a structural equation model (SEM) that shares the same but unknown directed acyclic graph (DAG) $\mathcal{G}$. The data-generating process can be summarized as follows:

$$\text{(Mixing)} \quad X = g(Z),$$

$$\text{(SEM)} \quad Z_i = f_i(\text{PA}(Z_i; \mathcal{G}), \epsilon_i; \theta_i^{(u)}), \ i \in [n], \quad (1)$$

$$Y = f_Y(\text{PA}(Y; \mathcal{G}), \epsilon_Y; \theta_Y^{(u)}).$$

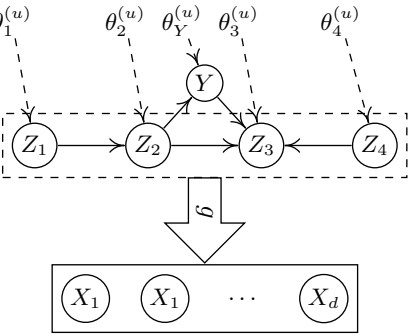

Figure 1: An example of the generative process considered in our work. The feature vector $X$ is generated from latent variables $Z$, which, along with the label $Y$, follow a structural equation model. The causal mechanisms, governed by parameters $\theta_i^{(u)}$ and $\theta_Y^{(u)}$, may shift across domains. Here, $X$ and the domain index $u$ are observable. Furthermore, the label $Y$ is available in the source domains but remains unobserved in the target domain. For this example, we have $Z_{\text{mb}} = \{Z_2, Z_3, Z_4\}$, $Z_{\text{pa}} = \{Z_2\}$, $Z_{\text{ch}} = \{Z_3\}$, $Z_{\text{sps}} = \{Z_4\}$, and $Z_{\text{mb}}^{\complement} = \{Z_1\}$. To illustrate these latent variables, consider an example from PACS benchmark (Li et al., 2017): $Y$ represents whether it is a horse, while $Z_2$ captures key defining features (e.g., a horse's head or horseshoes), and $Z_3$ represents attributes influenced by the horse (e.g., a saddle). Meanwhile, $Z_1$ and $Z_4$ can represent background elements.

Here, $\text{PA}(Z_i; \mathcal{G})$ and $\text{PA}(Y; \mathcal{G})$ represent the parents of $Z_i$ and $Y$, respectively, in the DAG $\mathcal{G}$. The $\epsilon_i$'s are mutually independent exogenous noise variables, and $\theta_i^{(u)}$ denotes the effective parameters (or latent factors) associated with each structural equation in the $u$-th domain. The generative process of each latent variable $Z_i$ may vary across domains, with the variation being determined by the corresponding parameters $\theta_i^{(u)}$. Such variability is common in practice, e.g., arising from heterogeneous datasets, where the causal mechanisms may shift. An example of the generative process is depicted in Figure 1.

Let $P_{X,Y}(X, Y; \theta^{(u)})$ and $P_{Z,Y}(Z, Y; \theta^{(u)})$ represent the joint distributions of $X, Y$ and $Z, Y$, respectively, in the $u$-th domain. When the context is clear, we omit the subscript for simplicity, and write $P^{(u)}(X, Y)$ and $P^{(u)}(Z, Y)$, respectively. We also assume that $P_{Z,Y}$ and $\mathcal{G}$ satisfy the faithfulness assumption (Spirtes et al., 2001), and that $P_Z$ is third-order differentiable and positive everywhere on $\mathbb{R}^n$.

Furthermore, we denote by $Z_{\text{mb}}$, $Z_{\text{pa}}$, $Z_{\text{ch}}$, and $Z_{\text{sps}}$ the Markov blanket[1], parents, children, and spouses of label $Y$, respectively. Also, let $Z_{\text{mb}}^{\complement}$ denote the remaining latent

---

[1]In this work, we use the term "Markov blanket" to refer to the parents, children, and spouses of a target variable.

variables outside the Markov blanket of $Y$, and $\mathcal{M}$ be the Markov network over latent variables $Z$ and label $Y$, whose edges are denoted by $\mathcal{E}(\mathcal{M})$. We define $\theta_{\mathrm{mb}} = (\theta_i)_{Z_i \in Z_{\mathrm{mb}}}$, and similarly for $\theta_{\mathrm{pa}}$, $\theta_{\mathrm{ch}}$, and $\theta_{\mathrm{sps}}$. We also denote by $\hat{Z}$, $\hat{\mathcal{G}}$, and $\hat{\mathcal{M}}$ the learned latent variables, learned DAG, and learned Markov networks, respectively.

## 4. Identifiability Theory

We present the identifiability theory for domain adaptation in a universal setting, where changes are allowed to occur anywhere in the latent space without restrictions. It is worth noting that specific types of domain shifts can be captured by imposing constraints on where changes occur. For instance:

- Restricting changes to the parents of $Y$ can be viewed as the covariate shift setting (Shimodaira, 2000).

- Restricting changes to $Y$ itself can be viewed as the target shift (Zhang et al., 2013) or prior probability shift (Storkey, 2009) problem.

- Restricting changes to the children of $Y$ can be viewed as the conditional shift (Zhang et al., 2013) problem.

In contrast, our work considers the most general scenario, where changes may occur anywhere in the latent space, without imposing any specific restrictions.

In Section 4.1, we discuss how learning latent representations of the label's Markov blanket enables adaptation to certain types of domain shifts, while highlighting why this approach is often insufficient for domain adaptation. We then propose an alternative approach in Section 4.2 that involves learning latent representations of the label's parents, children, and spouses. Finally, in Section 4.3, we provide the identifiability guarantee for this approach.

### 4.1. Subspace Identifiability of Latent Representations for Label's Markov Blanket

With the advent of deep learning and its widespread adoption, many approaches leverage deep learning to learn compact latent representations of observations (Ganin & Lempitsky, 2015). These representations facilitate knowledge transfer in the latent space, enabling more efficient and effective transfer. The critical goal is then to learn latent representations that retain predictive information about $Y$. A traditional view is that the Markov blanket contains all information sufficient for predicting the target variable. Building on this perspective, a natural approach is to learn representations that correspond to the Markov blanket of the label $Y$.

More specifically, the aim is to learn a representation $\hat{Z}_{\mathrm{mb}}$ that is an invertible transformation of the label's Markov

blanket $Z_{\mathrm{mb}}$, ensuring that $\hat{Z}_{\mathrm{mb}}$ contains all and only the information in $Z_{\mathrm{mb}}$. If such a representation can be recovered, we say that $Z_{\mathrm{mb}}$ is *subspace identifiable*. With such a representation, the label $Y$ becomes conditionally independent of all other variables $Z_{\mathrm{mb}}^{\complement}$, given $\hat{Z}_{\mathrm{mb}}$. This implies that $\hat{Z}_{\mathrm{mb}}$ captures all essential information required to predict $Y$. Notably, this approach aligns with the feature selection literature (Yu et al., 2020), where the Markov blanket is recognized as the minimal predictive set for the target variable.

However, recovering such a Markov blanket representation $\hat{Z}_{\mathrm{mb}}$ is challenging without additional assumptions, as latent variable modeling often admits many spurious solutions (Hyvärinen & Pajunen, 1999; Locatello et al., 2019). Fortunately, access to multi-domain data makes this recovery feasible. To achieve this, we rely on specific assumptions that require the distribution of latent variables to vary sufficiently across the source domains, formally described below.

**Assumption 1** (Sufficient changes for $Z$). *For each value of $Z$, there exist $2n + |\mathcal{M}| + 1$ values of $u$, i.e., $u_k$ with $k = 0, \ldots, 2n + |\mathcal{M}|$, such that the vectors $w(Z, u_k) - w(Z, u_0)$ with $k = 1, \ldots, 2n + |\mathcal{M}|$ are linearly independent, where vector $w(Z, u)$ is defined as*

$$w(Z, u) = \left( \frac{\partial \log P^{(u)}(Z, Y)}{\partial Z_i} \right)_{i \in [n]}$$
$$\oplus \left( \frac{\partial^2 \log P^{(u)}(Z, Y)}{\partial Z_i^2} \right)_{i \in [n]}$$
$$\oplus \left( \frac{\partial^2 \log P^{(u)}(Z, Y)}{\partial Z_i \partial Z_j} \right)_{\{Z_i, Z_j\} \in \mathcal{E}(\mathcal{M}), i < j}.$$

**Assumption 2** (Sufficient changes for $Y$). *For each value of $Z$, there exist $|Z_{\mathrm{mb}}| + 1$ values of $(u, c)$ such that the vectors $\tau(Z, u_k, c_r) - \tau(Z, u_k, c_1)$ with $c_r \neq c_1$ are linearly independent, where vector $\tau(Z, u, c)$ is defined as*

$$\tau(Z, u, c) = \left( \frac{\partial \log P^{(u)}(Z, Y = v_c)}{\partial Z_i} \right)_{Z_i \in Z_{\mathrm{mb}}}.$$

It is worth noting that different forms of sufficient change conditions have been adopted in nonlinear ICA (Hyvärinen et al., 2023) and causal representation learning (Schölkopf et al., 2021). These distribution changes, along with the invariant mixing function, offer valuable information for inferring the latent variables and their relations. We now provide identifiability theory to learn the latent representations for the label's Markov blanket. The proof is provided in Appendix A and is inspired by Zhang et al. (2024). Although we state the faithfulness assumption (Spirtes et al., 2001) in the theorem above and Theorem 2, it suffices to adopt the single adjacency-faithfulness (SAF) and single unshielded-collider-faithfulness (SUCF) assumptions (Ng et al., 2021; Zhang et al., 2024). These assumptions are

considerably weaker than the faithfulness assumption and ensure that the Markov network $\mathcal{M}$ is the same as the moralized graph of the DAG $\mathcal{G}$ (Zhang et al., 2024, Proposition 2).

**Theorem 1** (Subspace identifiability of Markov blanket). *Consider the generative process in Equation* (1). *Suppose that Assumptions 1 and 2, as well as the faithfulness assumption, hold. By modeling the same generative process with minimal number of edges for the learned Markov network $\hat{\mathcal{M}}$, the learned Markov blanket $\hat{Z}_{\mathrm{mb}}$ is an invertible transformation of the true Markov blanket $Z_{\mathrm{mb}}$.*

However, learning latent representations that correspond to the subspace of the label's Markov blanket is insufficient for domain adaptation in many scenarios. For instance, consider the factorization of the joint distribution $P(Z_{\mathrm{mb}}, Y) = P(Y \mid Z_{\mathrm{mb}})P(Z_{\mathrm{mb}})$. If $P(Z_{\mathrm{mb}})$ changes across domains while $P(Y \mid Z_{\mathrm{mb}})$ remains invariant, domain adaptation can be achieved by using the same classifier (with $Z_{\mathrm{mb}}$ or $\hat{Z}_{\mathrm{mb}}$ as input) trained on the source domains in the target domain. This corresponds to a scenario where the conditional distributions $P(Z_{\mathrm{pa}} \mid \mathrm{PA}(Z_{\mathrm{pa}}; \mathcal{G}))$ or $P(Z_{\mathrm{sps}} \mid \mathrm{PA}(Z_{\mathrm{sps}}; \mathcal{G}))$ change across domains, while $P(Y \mid Z_{\mathrm{ch}})$ and $P(Z_{\mathrm{ch}} \mid \mathrm{PA}(Z_{\mathrm{ch}}; \mathcal{G}))$ remain invariant, which is clearly restrictive.

Now consider an alternative scenario where the factorization is given by $P(Z_{\mathrm{mb}}, Y) = P(Z_{\mathrm{mb}} \mid Y)P(Y)$, where $P(Z_{\mathrm{mb}} \mid Y)$ remains invariant across domains while $P(Y)$ changes. This is known as the target shift (Zhang et al., 2013) or prior probability shift (Storkey, 2009) problem. However, with subspace identifiability of the label's Markov blanket indicated by Theorem 1, we do not know which parts of the learned representations correspond to the label's children, spouses, or parents. In this case, one may also factorize the distribution as $P(Z_{\mathrm{mb}}, Y) = P(Y \mid Z_{\mathrm{mb}})P(Z_{\mathrm{mb}})$, where both conditional distributions are allowed to change. Since $Y$ is not available in the target domain and $P(Y \mid Z_{\mathrm{mb}})$ changes, one no longer has identifiability of distribution $P(Z_{\mathrm{mb}}, Y)$ in the target domain.

This motivates us to separate the representations of $Z_{\mathrm{mb}}$ into three different subspaces in the next subsection, allowing us to improve the identifiability and to have a more parsimonious representation of the changes.

## 4.2. Subspace Identifiability of Latent Representations for Label's Parents, Children, and Spouses

In the previous subsection, we demonstrated that learning latent representations corresponding to the subspace of the label's Markov blanket is often insufficient for domain adaptation. This limitation arises, in part, because such representations are overly coarse-grained. To address this issue, we propose a more fine-grained approach that involves learn-

ing latent representations corresponding to three distinct subspaces of the label's Markov blanket: its parents, children, and spouses. Conceptually, this can be viewed as partitioning the Markov blanket into these three subspaces and focusing on recovering each subspace separately. In Section 4.3, we will show how such representations enable domain adaptation with identifiability guarantee in a universal setting.

Before presenting the assumptions and identifiability theory, we first introduce the notion of an *intimate neighbor*. Specifically, a latent variable $Z_i$ is said to be an intimate neighbor of $Z_j$ if $Z_i$ is adjacent to $Z_j$ and to all other neighbors of $Z_j$ in $\mathcal{M}$. Based on this, we introduce the following structural assumption on the latent DAG $\mathcal{G}$:

**Assumption 3** (Group-specific intimate neighbors). *The intimate neighbors of label $Y$'s parents, children, and spouses can only have intimate neighbors—excluding $Y$ itself—within their respective groups, i.e., other parents, children, or spouses of $Y$.*

This assumption is rather mild as it permits edges among the parents, children, and spouses of $Y$, but restricts edges between intimate neighbors belonging to different groups. In practice, intimate neighbors may be relatively rare. This assumption is necessary because, without additional conditions, it is generally not possible to disentangle $Z_i$ from $Z_j$ if $Z_i$ is an intimate neighbor of $Z_j$, as supported by the theory in Zhang et al. (2024).

Next, we present the identifiability theory for learning representations of the subspaces corresponding to parents, children, and spouses. The proof is given in Appendix B.

**Theorem 2** (Subspace identifiability of parents, children, and spouses). *Consider the generative process in Equation* (1). *Suppose that Assumptions 1, 2 and 3, as well as the faithfulness assumption, hold. By modeling the same generative process with minimal number of edges for the learned Markov network $\hat{\mathcal{M}}$, there exists a partition of the learned Markov blanket $\hat{Z}_{\mathrm{mb}}$, denoted as $\hat{Z}_{S_1}$, $\hat{Z}_{S_2}$, and $\hat{Z}_{S_3}$, such that they are invertible transformations of the true parents $Z_{\mathrm{pa}}$, children $Z_{\mathrm{ch}}$, and spouses $Z_{\mathrm{sps}}$, respectively.*

The above theorem implies that the latent representations of parents, children, and spouses can be disentangled, allowing for the recovery of their respective subspaces. In the next subsection, we explain how these representations facilitate domain adaptation in a universal setting with identifiability guarantee.

## 4.3. Identifiability of Joint Distribution in Target Domain

Building on the identifiability of latent representations established in Section 4.2, we now demonstrate how this facilitates domain adaptation with identifiability guarantees in a

general setting. Specifically, the objective is to identify the joint distribution $P^\tau(X, Y)$ in the unlabeled target domain, or equivalently, $P^\tau(Y \mid X)$, since $P^\tau(X)$ is already known in the target domain.

To relate the conditional distribution $P^\tau(Y \mid X)$ at the level of raw observations $X$ to the latent representations, we first state the following proposition and provide the proof in Appendix C.1.

**Proposition 1.** *Consider the generative process in Equation* (1). *We have*

$$P^\tau(Y = v_k \mid X)$$
$$= \frac{P^\tau(Z_{\mathrm{ch}} \mid Y = v_k, Z_{\mathrm{sps}}) P^\tau(Y = v_k \mid Z_{\mathrm{pa}})}{\sum_{c=1}^C P^\tau(Z_{\mathrm{ch}} \mid Y = v_c, Z_{\mathrm{sps}}) P^\tau(Y = v_c \mid Z_{\mathrm{pa}})}.$$

The above proposition implies that, to identify $P^\tau(Y \mid X)$, it suffices to identify $P^\tau(Z_{\mathrm{ch}} \mid Y = v_c, Z_{\mathrm{sps}})$ and $P^\tau(Y \mid Z_{\mathrm{pa}})$ in the target domain. These conditional distributions are often simpler to model. However, the underlying latent variables $Z_{\mathrm{pa}}$, $Z_{\mathrm{ch}}$, and $Z_{\mathrm{sps}}$ are not directly observable and cannot be exactly recovered.

Fortunately, the identifiability theory developed in Section 4.2 shows that the subspaces corresponding to latent variables $Z_{\mathrm{pa}}$, $Z_{\mathrm{ch}}$, and $Z_{\mathrm{sps}}$ can be disentangled. Specifically, if one can learn representations $\hat{Z}_{\mathrm{pa}}$, $\hat{Z}_{\mathrm{ch}}$, and $\hat{Z}_{\mathrm{sps}}$ that are invertible transformations of $Z_{\mathrm{pa}}$, $Z_{\mathrm{ch}}$, and $Z_{\mathrm{sps}}$, respectively, then we have the following result.

**Corollary 1.** *Consider the generative process in Equation* (1). *Let* $\hat{Z}_{\mathrm{pa}}$, $\hat{Z}_{\mathrm{ch}}$, *and* $\hat{Z}_{\mathrm{sps}}$ *be invertible transformations of* $Z_{\mathrm{pa}}$, $Z_{\mathrm{ch}}$, *and* $Z_{\mathrm{sps}}$, *respectively. We have*

$$P^\tau(Y = v_k \mid X)$$
$$= \frac{P^\tau(\hat{Z}_{\mathrm{ch}} \mid Y = v_k, \hat{Z}_{\mathrm{sps}}) P^\tau(Y = v_k \mid \hat{Z}_{\mathrm{pa}})}{\sum_{c=1}^C P^\tau(\hat{Z}_{\mathrm{ch}} \mid Y = v_c, \hat{Z}_{\mathrm{sps}}) P^\tau(Y = v_c \mid \hat{Z}_{\mathrm{pa}})}.$$

The proof is available in Appendix C.2. From the corollary above, it suffices to identify the subspaces of the latent variables $Z_{\mathrm{pa}}$, $Z_{\mathrm{ch}}$, and $Z_{\mathrm{sps}}$ up to invertible transformations, e.g., by leveraging the identifiability theory developed in Section 4.2. Furthermore, the corollary implies that it suffices to establish the identifiability of the conditional distributions $P^\tau(\hat{Z}_{\mathrm{ch}} \mid Y, \hat{Z}_{\mathrm{sps}})$ and $P^\tau(Y \mid \hat{Z}_{\mathrm{pa}})$ in the target domain.

To ensure identifiability, we adopt the minimal change principle, which posits that the distributional changes across domains are confined to a low-dimensional manifold (Stojanov et al., 2019). Specifically, we assume that the conditional distributions are governed by a small number of identifiable changing parameters, inspired by Stojanov et al. (2019). This enables us to identify these parameters by learning low-dimensional representations of the conditional distributions that vary across the source domains.

**Assumption 4** (Low-dimensional changes). *For each value of* $v_c$, *the conditional distribution* $P(Z_{\mathrm{ch}} \mid Y = v_c, Z_{\mathrm{sps}})$ *contains only a finite number of identifiable parameters that vary across domains. Furthermore, there is a sufficiently large number of source domains.*

Similar to Stojanov et al. (2019), Assumption 4 implies the existence of a bijective transformation $h : \mathcal{P}_{Z_{\mathrm{ch}} \mid \mathcal{Y}, Z_{\mathrm{sps}}} \to \mathbb{R}^q$, where $q$ denotes the dimensionality of the effective changing parameters. Under this transformation, the conditional distribution in each domain $u$ can be expressed as a linear combination of the conditional distributions in the other source domains, i.e., $h\big(P_{Z_{\mathrm{ch}} \mid Y = v_c, Z_{\mathrm{sps}}}^{(u)}\big) = \sum_{i=1, i \neq u}^M \alpha_{ic}^{(u)} h\big(P_{Z_{\mathrm{ch}} \mid Y = v_c, Z_{\mathrm{sps}}}^{(i)}\big)$ for some mixture weights $\alpha_{1c}^{(u)}, \dots, \alpha_{Mc}^{(u)}$. Similarly, for the target domain $\tau$, there exist weights $\alpha_{1c}^\tau, \dots, \alpha_{Mc}^\tau$ such that $h\big(P_{Z_{\mathrm{ch}} \mid Y = v_c, Z_{\mathrm{sps}}}^\tau\big) = \sum_{i=1}^M \alpha_{ic}^\tau h\big(P_{Z_{\mathrm{ch}} \mid Y = v_c, Z_{\mathrm{sps}}}^{(i)}\big)$. More intuitively, Assumption 4 indicates that all domain-specific conditional distributions (including source and target domains) for the label $v_c$ are confined to a $q$-dimensional manifold. Therefore, each conditional distribution for domain $u$ can be characterized by the mixture weights $\alpha_{1c}^{(u)}, \dots, \alpha_{Mc}^{(u)}$. We denote the conditional distribution associated with weights $\alpha_c$ as $P^{\alpha_c}(Z_{\mathrm{ch}} \mid Y = v_c, Z_{\mathrm{sps}})$.

We also adopt the following assumption, which ensures that the changes in conditional distributions are linearly independent. This is a rather mild assumption which requires that the conditional distribution varies sufficiently when their parameters change; otherwise, such parameter changes will not leave a sufficient footprint on the distribution shifts.

**Assumption 5** (Linear independence). *The elements in the set* $\{\beta_c P^{\alpha_c}(Z_{\mathrm{ch}} \mid Y = v_c, Z_{\mathrm{sps}}) + \beta_c' P^{\alpha_c'}(Z_{\mathrm{ch}} \mid Y = v_c, Z_{\mathrm{sps}}); c = 1, \dots, C\}$ *are linearly independent for all* $\alpha_c, \alpha_c', \beta_c, \beta_c', \beta_c + \beta_c' \neq 0$, *if they are not zero.*

With the assumptions above, we provide the identifiability result for the conditional distributions in the target domain.

**Theorem 3** (Identifiability of target distribution). *Suppose that Assumptions 4 and 5 hold. Let* $\hat{Z}_{\mathrm{pa}}$, $\hat{Z}_{\mathrm{ch}}$, *and* $\hat{Z}_{\mathrm{sps}}$ *be invertible transformations of* $Z_{\mathrm{pa}}$, $Z_{\mathrm{ch}}$, *and* $Z_{\mathrm{sps}}$, *respectively. Suppose that we learn* $P^{new}$ *to match* $P^\tau(\hat{Z}_{\mathrm{ch}} \mid \hat{Z}_{\mathrm{pa}}, \hat{Z}_{\mathrm{sps}})$ *in the target domain, i.e.,* $P^{new}(\hat{Z}_{\mathrm{ch}} \mid \hat{Z}_{\mathrm{pa}}, \hat{Z}_{\mathrm{sps}}) = P^\tau(\hat{Z}_{\mathrm{ch}} \mid \hat{Z}_{\mathrm{pa}}, \hat{Z}_{\mathrm{sps}})$ *while constraining* $P^{new}(\hat{Z}_{\mathrm{ch}} \mid Y, \hat{Z}_{\mathrm{sps}})$ *to satisfy Assumption 4. Then, we have* $P^\tau(\hat{Z}_{\mathrm{ch}} \mid Y, \hat{Z}_{\mathrm{sps}}) = P^{new}(\hat{Z}_{\mathrm{ch}} \mid Y, \hat{Z}_{\mathrm{sps}})$ *and* $P^\tau(Y \mid \hat{Z}_{\mathrm{pa}}) = P^{new}(Y \mid \hat{Z}_{\mathrm{pa}})$.

The proof is given in Appendix D and is inspired by Stojanov et al. (2019). The core idea is that the learned low-dimensional representations allow us to reconstruct the conditional distribution in the target domain using unlabeled data in the target domain. Combined with the linear independence assumption, this further facilitates label prediction

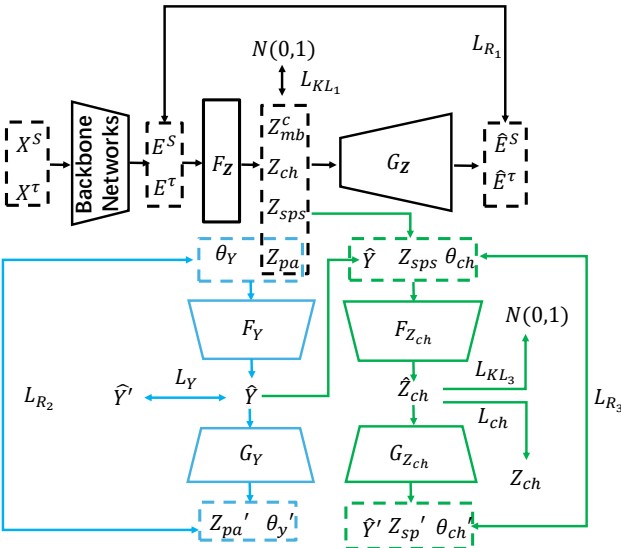

Figure 2: **Overview of the General Approach for Multi-source Domain Adaptation (GAMA).** The model first maps input images $X$ to a latent space $Z$ using a VAE framework. The latent variables $Z$ are partitioned into several components: $Z_{\mathrm{mb}}$, $Z_{\mathrm{pa}}$, $Z_{\mathrm{ch}}$, and $Z_{\mathrm{sps}}$. Two additional VAEs are employed to capture the relationships among the latent variables and label, which aids in estimating $\theta$ for improved predictions. For the three VAEs in total, we have the following losses: $\mathcal{L}_{\mathrm{vae},Z} = \mathcal{L}_{\mathrm{KL}_1} + \mathcal{L}_{\mathrm{R}_1}$, $\mathcal{L}_{\mathrm{vae},Y} = \mathcal{L}_{\mathrm{R}_2}$ and $\mathcal{L}_{\mathrm{vae},Z_{ch}} = \mathcal{L}_{\mathrm{KL}_3} + \mathcal{L}_{\mathrm{R}_3}$. Cross-entropy loss $\mathcal{L}_Y$ and mean squared error (MSE) loss $\mathcal{L}_{\mathrm{ch}}$ are also used in the source domains to encourage better encoding. The final prediction is made by training a classifier on the inputs $\left(Z_{\mathrm{pa}}, Z_{\mathrm{sps}}, Z_{\mathrm{ch}}, \theta_Y^{(u)}, \theta_{\mathrm{ch}}^{(u)}\right)$.

in the target domain. It is worth noting that the result can be straightforwardly extended to multi-target domain adaptation by learning distinct $P^{\mathrm{new}}$ for each target domain.

**Remark 1.** *In summary, one can first utilize Theorem 2 to learn a demixing function $\hat{g}^{-1}$ (i.e., an encoder) that extracts latent representations of the label's parents, children, and spouses, up to certain indeterminacies. This same demixing function can then be applied to the target domain, where Theorem 3 guarantees the identifiability of the distributions $P^{\tau}(\hat{Z}_{\mathrm{ch}} \mid Y, \hat{Z}_{\mathrm{sps}})$ and $P^{\tau}(Y \mid \hat{Z}_{\mathrm{pa}})$ in the target domain. Finally, applying Corollary 1 ensures the identifiability of $P^{\tau}(Y \mid X)$ in the target domain.*

## 5. Domain Adaptation Approach

Building on the theoretical insights established in the previous section, we propose a **G**eneral **A**pproach for **M**ulti-source domain **A**daptation (**GAMA**) that systematically learns and identifies both latent variable structures and label information in all domains. Our approach incorporates

representation learning to characterize distributional shifts across domains, drawing inspiration partly from the framework presented by Zhang et al. (2020). The approach operates through a principled multi-stage process grounded in identifiability theory and the necessity of isolating different components in the Markov blanket for accurate prediction of the target variable $Y$.

First, we use variational autoencoders (VAEs) to match the distributions across source and target domains, extracting the required latent representations. Subsequently, we employ additional VAE (Kingma & Welling, 2014) modules to explicitly model inter-variable dependencies within the latent space, enabling systematic decomposition into block-level components while simultaneously estimating domain-specific parameters $\theta$. This design ensures that our framework effectively captures all variables constituting the Markov blanket of $Y$, thereby facilitating robust cross-domain generalization and achieving accurate predictions in the target domain. Note that we use VAEs because they provide a convenient way to model the distribution of latent variables, and make it easier to incorporate prior structural information (e.g., parent-child relationships) into our method.

We now describe the specific model architecture. We first take an input image $X$ and pass it through a backbone network (e.g., ResNet-50 (He et al., 2016)) to obtain a feature representation $E$. An encoder $F_Z$ then maps $E$ into a latent space $Z$. Since we adopt a VAE framework (Kingma & Welling, 2014), a decoder $G_Z$ is also introduced to reconstruct $E$ from $Z$. The reconstruction loss from the VAE enforces consistency between the original feature representation $E$ and its reconstructed version, preserving essential information. Here, we have the loss $\mathcal{L}_{\mathrm{vae},Z} = \mathcal{L}_{\mathrm{KL}_1} + \mathcal{L}_{\mathrm{R}_1}$. Note that $\mathcal{L}_{\mathrm{R}}$ denotes reconstruction loss, while $\mathcal{L}_{\mathrm{KL}}$ denotes Kullback–Leibler (KL) divergence; the index indicates loss for different VAEs. For example, $\mathcal{L}_{\mathrm{KL}_1}$ denotes the KL divergence of the VAE from $X$ to $Z$.

We partition the latent variable $Z$ as

$$Z = \left(Z_{\mathrm{mb}}^{\complement}, Z_{\mathrm{pa}}, Z_{\mathrm{ch}}, Z_{\mathrm{sps}}\right) \in \mathbb{R}^n.$$

According to Figure 1, we observe that, given $Z_{\mathrm{mb}}$, the elements relevant to $Y$ still include $\theta_Y^{(u)}$ and $\theta_{\mathrm{ch}}^{(u)}$. Once $\theta_Y^{(u)}$ and $\theta_{\mathrm{ch}}^{(u)}$ are accurately identified, combining them with $Z_{\mathrm{mb}}$ yields a stable prediction (since all relevant information is then obtained).

Consider the data generation process involving $\theta_Y^{(u)}$ and $\theta_{\mathrm{ch}}^{(u)}$:

$$\left(\theta_Y^{(u)}, Z_{\mathrm{pa}}\right) \mapsto Y \quad \text{and} \quad \left(\theta_{\mathrm{ch}}^{(u)}, Y, Z_{\mathrm{sps}}\right) \mapsto Z_{\mathrm{ch}}.$$

In each domain, these $\theta$ values are fixed parameters. Thus, we aim to learn $\theta$ and $Y$ so as to maximize $P(Z \mid \theta)$ and

$P(Z, Y \mid \theta)$ in the target domain. Formally, it is given by

$$
\max_{\theta_Y^{(u)}, \theta_{\mathrm{ch}}^{(u)}, q_1, q_2} \Big( \mathbb{E}_{Y \sim q_1(Y \mid Z_{\mathrm{pa}}, \theta_Y^{(u)})} \log p_1(Z_{\mathrm{pa}}, \theta_Y^{(u)} \mid Y)
$$
$$
- \beta_1 \, \mathrm{KL}\big( q_1(Y \mid Z_{\mathrm{pa}}, \theta_Y^{(u)}) \parallel P(Y) \big)
$$
$$
+ \mathbb{E}_{Z_{\mathrm{ch}} \sim q_2(Z_{\mathrm{ch}} \mid Z_{\mathrm{sps}}, Y, \theta_{\mathrm{ch}}^{(u)})} \log p_2(Z_{\mathrm{sps}}, Y, \theta_{\mathrm{ch}}^{(u)} \mid Z_{\mathrm{ch}})
$$
$$
- \beta_2 \, \mathrm{KL}\big( q_2(Z_{\mathrm{ch}} \mid Z_{\mathrm{sps}}, Y, \theta_{\mathrm{ch}}^{(u)}) \parallel p(Z_{\mathrm{ch}}) \big) \Big).
$$

Two VAEs are used here. Specifically, $(\theta_Y^{(u)}, Z_{\mathrm{pa}}) \mapsto Y$ involves an encoder $F_Y$ and a decoder $G_Y$, while $(\theta_{\mathrm{ch}}^{(u)}, Y, Z_{\mathrm{sps}}) \mapsto Z_{\mathrm{ch}}$ involves an encoder $F_{Z_{\mathrm{ch}}}$ and a decoder $G_{Z_{\mathrm{ch}}}$. We set $\beta_1$ and $\beta_2$ to 1. These lead to the losses $\mathcal{L}_{\mathrm{vae},Y}$ and $\mathcal{L}_{\mathrm{vae},Z_{\mathrm{ch}}}$. Note that since $Y$ is discrete, we cannot assume that $P(Y)$ is a Gaussian distribution (which is commonly done in VAE estimation), and thus we use a Gumbel-Softmax VAE (Jang et al., 2017) which can convert the logit of $Y$ into continuous variables for further calculation . In the training stage, we treat $F_Y$ as the encoder producing $\hat{Y}$. Since we have access to the ground truth labels $Y$ in the source domains, we simply calculate the cross-entropy between $Y$ and $\hat{Y}$, giving rise to the losses $\mathcal{L}_Y$ and $\mathcal{L}_{\mathrm{vae},Y} = \mathcal{L}_{\mathrm{R}_2}$.

Furthermore, in the source domains, since we have access to the ground truth $Z_{\mathrm{ch}}$, we have the following MSE loss based on the encoded values $\hat{Z}_{\mathrm{ch}}$ to better capture the relationships between variables and the ground truth: $\mathcal{L}_{\mathrm{ch}} = \mathrm{MSE}(Z_{\mathrm{ch}}, \hat{Z}_{\mathrm{ch}})$, where $\mathrm{MSE}(\cdot)$ denotes the mean squared error. Also, for the other VAE, we have $\mathcal{L}_{\mathrm{vae},Z_{\mathrm{ch}}} = \mathcal{L}_{\mathrm{KL}_3} + \mathcal{L}_{\mathrm{R}_3}$.

Finally, we can make the final prediction by using $(Z_{\mathrm{pa}}, Z_{\mathrm{sps}}, Z_{\mathrm{ch}}, \theta_Y^{(u)}, \theta_{\mathrm{ch}}^{(u)})$ with loss $\mathcal{L}_{\mathrm{cls}}$. In conclusion, we have the following loss during training, where $\lambda_1, \lambda_2, \lambda_3, \lambda_4$ and $\lambda_5$ are hyperparameters:

$$
\mathcal{L}_{\mathrm{all}} = \mathcal{L}_{\mathrm{cls}} + \lambda_1 \mathcal{L}_{\mathrm{vae},Z} + \lambda_2 \mathcal{L}_{\mathrm{vae},Y}
$$
$$
+ \lambda_3 \mathcal{L}_{\mathrm{vae},Z_{\mathrm{ch}}} + \lambda_4 \mathcal{L}_{\mathrm{ch}} + \lambda_5 \mathcal{L}_Y.
$$

# 6. Experiments

We show the effectiveness of our method compared with existing ones on widely used datasets in domain adaptation. Further details and empirical studies can be found in Appendix E.

## 6.1. Datasets and Baselines

**Datasets.** We validate our method on two well-known benchmarks for domain adaptation: Office-Home (Venkateswara et al., 2017) and PACS (Li et al., 2017). In each dataset, a single domain is designated as the target, and the remaining domains serve as sources. For Office-Home, we extract features using a pretrained ResNet50,

then apply MLP-based VAEs alongside a classifier. Meanwhile, for PACS, we employ ResNet18 as the backbone and similarly integrate MLP-based VAEs and a classifier. All metrics are computed by averaging over three random seeds.

**Baselines.** To assess performance, we compare against several baselines, including the Source Only (He et al., 2016) approach and single-source domain adaptation methods such as DAN (Long et al., 2015), MCD (Saito et al., 2018), and DANN+BSP (Chen et al., 2019c). We further evaluate our model against leading multi-source domain adaptation techniques, including M3SDA (Peng et al., 2019), CMSS (Yang et al., 2020), LtC-MSDA (Wang et al., 2020), and T-SVDNet (Li et al., 2021). Additionally, we incorporate comparisons with WADN (Shui et al., 2021), which handles target shift in multi-source scenarios, as well as iMSDA (Kong et al., 2022), a recent framework that leverages component-wise identification for MSDA.

## 6.2. Numerical Results

The results for Office-Home and PACS datasets are provided in Table 1.

**Office-Home dataset.** **GAMA** achieves the best performance in most sub-tasks. On average, **GAMA** surpasses the strongest baseline (iMSDA) by a margin of 1%. This is because our model tries to learn the pattern in the target domain while training. By effectively predicting $\theta$, our method is able to make more accurate inferences, leading to better performance.

**PACS dataset.** **GAMA** performs well for this dataset and achieves better accuracy than the best baseline on average. In particular, in the Photo domain, where the accuracy is already high, we have achieved an accuracy of 98.8%, which means that we have further explored the potential of the data.

## 6.3. Ablation Study

To evaluate the effectiveness of our special design to capture $\theta$ in the target domain, we design two model variants: (1) GAMA-vae: we remove all the VAE related losses; (2) GAMA-theta: we remove the losses brought by $\theta$-related VAEs: $\mathcal{L}_{\mathrm{vae},Y}, \mathcal{L}_{\mathrm{vae},Z_{\mathrm{ch}}}$ , we also remove $\mathcal{L}_{\mathrm{ch}}$, and $\mathcal{L}_Y$ as some items for calculating these losses are related to $\theta$. Experiment results on the Office-Home dataset are shown in Table 2. It shows that the VAEs are essential for capturing information to perform adaptation. Moreover, with the $\theta$-related VAEs, one observes an improved accuracy.

# 7. Conclusion

We develop a general, representation-based domain adaptation framework that can handle different types of distribution

Table 1: Results on Office-Home (Ar, Cl, Pr, Rw) and PACS (P, A, C, S). A dash "-" indicates no reported result. Baseline results are taken from Kong et al. (2022).

| Method | Office-Home | | | | | PACS | | | | |
|---|---|---|---|---|---|---|---|---|---|---|
| | Ar | Cl | Pr | Rw | Avg | P | A | C | S | Avg |
| DAN (Long et al., 2015) | 68.3 | 57.9 | 78.5 | 81.9 | 71.6 | - | - | - | - | - |
| Source Only (He et al., 2016) | 64.6 | 52.3 | 77.6 | 80.7 | 68.8 | 94.5 | 74.9 | 72.1 | 64.7 | 76.6 |
| DANN (Ganin et al., 2016) | 64.3 | 58.0 | 76.4 | 78.8 | 69.4 | 91.8 | 81.9 | 77.5 | 74.6 | 81.5 |
| DCTN (Xu et al., 2018) | 66.9 | 61.8 | 79.2 | 77.8 | 71.4 | - | - | - | - | - |
| MDAN (Zhao et al., 2018) | - | - | - | - | - | 91.4 | 79.1 | 76.0 | 72.0 | 79.6 |
| WBN (Mancini et al., 2018) | - | - | - | - | - | 97.4 | 89.9 | 89.7 | 58.0 | 83.8 |
| MCD (Saito et al., 2018) | 67.8 | 59.9 | 79.2 | 80.9 | 72.0 | 96.4 | 88.7 | 88.9 | 73.9 | 87.0 |
| DANN+BSP (Chen et al., 2019c) | 66.1 | 61.0 | 78.1 | 79.9 | 71.3 | - | - | - | - | - |
| M3SDA (Peng et al., 2019) | 66.2 | 58.6 | 79.5 | 81.4 | 71.4 | 97.3 | 89.3 | 89.9 | 76.7 | 88.3 |
| CMSS (Yang et al., 2020) | - | - | - | - | - | 96.9 | 88.6 | 90.4 | 82.0 | 89.5 |
| LtC-MSDA (Wang et al., 2020) | - | - | - | - | - | 97.2 | 90.2 | 90.5 | 81.5 | 89.8 |
| T-SVDNet (Li et al., 2021) | - | - | - | - | - | 98.5 | 90.4 | 90.6 | 85.5 | 91.3 |
| GeNRT (Deng et al., 2023) | - | - | - | - | - | 98.5 | 93.6 | 91.4 | 85.7 | 92.3 |
| iLCC-LCS (Liu et al., 2022) | - | - | - | - | - | 95.9 | 86.4 | 81.1 | 86.0 | 87.4 |
| WADN (Shui et al., 2021) | 75.2 | 61.0 | 83.5 | 84.4 | 76.1 | - | - | - | - | - |
| CASR (Wang et al., 2023) | 72.2 | 61.1 | 82.8 | 82.8 | 74.7 | - | - | - | - | - |
| TFFN (Li et al., 2023b) | 72.2 | 62.9 | 81.7 | 83.5 | 75.1 | - | - | - | - | - |
| SSD (Li et al., 2023a) | 72.5 | **64.5** | 81.2 | 83.2 | 75.4 | - | - | - | - | - |
| MIAN-$\gamma$ (Park & Lee, 2021) | 69.9 | 64.2 | 80.9 | 81.5 | 74.1 | - | - | - | - | - |
| iMSDA (Kong et al., 2022) | 75.4 | 61.4 | 83.5 | 84.5 | 76.2 | 98.5 | **93.8** | 92.5 | 89.2 | 93.5 |
| **GAMA (Ours)** | **76.6** | 62.6 | **84.9** | **84.9** | **77.3** | **98.8** | 93.7 | **92.8** | **89.3** | **93.7** |

Table 2: Ablation study on Office-Home comparing GAMA, GAMA-vae, and GAMA-theta.

| Method | Ar | Cl | Pr | Rw | Avg |
|---|---|---|---|---|---|
| GAMA | 76.6 | 62.6 | 84.9 | 84.9 | 77.3 |
| GAMA-vae | 74.9 | 60.5 | 83.4 | 84.8 | 75.9 |
| GAMA-theta | 75.3 | 61.7 | 83.4 | 84.8 | 76.0 |

shifts. Specifically, we show that learning subspace of the label's Markov blanket representations is often underspecified for domain adaptation in many scenarios. To achieve general domain adaptation, we show that one should partition the subspace of Markov blanket into the subspace of the label's parents, children, and spouses. We then establish identifiability of the joint distribution in the target domain. Our resulting method provides a practical solution to domain adaptation in general settings and outperforms existing methods on various benchmark datasets, highlighting its potential for broader applications. Future works include evaluating the method on larger-scale datasets and extending its application to diverse tasks, such as video, speech, and text.

## Impact Statement

Our paper presents a method for improving unsupervised domain adaptation to help AI models better transfer knowledge across different domains. The goal is to make models more efficient, especially in situations where labeled data is scarce. Our work does not introduce new ethical risks, as it focuses purely on enhancing the ability of AI systems to generalize across domains without affecting sensitive areas. This research aims to advance the field of machine learning, making it more practical and effective.

## Acknowledgments

The authors would like to thank the reviewers for their helpful comments. We would like to acknowledge the support from NSF Award No. 2229881, AI Institute for Societal Decision Making (AI-SDM), the National Institutes of Health (NIH) under Contract R01HL159805, and grants from Quris AI, Florin Court Capital, and MBZUAI-WIS Joint Program. IN acknowledges the support of the Natural Sciences and Engineering Research Council of Canada (NSERC) Postgraduate Scholarships – Doctoral program.

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

# Supplementary Material

## A. Proof of Theorem 1

To prove the following theorem, we begin by establishing several intermediate results that are useful. We first prove Proposition 2, which is used in the proof of Proposition 3. Building upon these two propositions, we then prove Proposition 4. Using Propositions 3 and 4, we proceed to establish Proposition 5. With these results, we are ready to prove the following theorem, leveraging Propositions 3 and 5. It is worth noting that the overall proof strategy is partly inspired by Zhang et al. (2024), while ours is considerably more complex as it involves the discrete target variable $Y$ (which is observed in the source domains).

**Theorem 1** (Subspace identifiability of Markov blanket). *Consider the generative process in Equation (1). Suppose that Assumptions 1 and 2, as well as the faithfulness assumption, hold. By modeling the same generative process with minimal number of edges for the learned Markov network $\hat{\mathcal{M}}$, the learned Markov blanket $\hat{Z}_{\mathrm{mb}}$ is an invertible transformation of the true Markov blanket $Z_{\mathrm{mb}}$.*

*Proof.* Recall that $\hat{Z}$ denotes the recovered latent variables, $\hat{\mathcal{M}}$ denotes the recovered Markov network, and $\Psi_{Z_i}$ denotes the intimate neighbors of $Z_i$. By Propositions 3 and 5, there exists a permutation $\pi$ of $\hat{Z}$, denoted as $\hat{Z}_\pi$, such that the following statements hold:

(a) $\hat{Z}_{\pi(i)}$ is solely a function of a subset of $\{Z_i\} \cup \Psi_{Z_i}$.

(b) $\hat{\mathcal{M}}_\pi$ and $\mathcal{M}$ are identical.

By Statement (b), under the faithfulness assumption (specifically the SAF and SUCF assumptions), the moralized graphs of $\hat{\mathcal{G}}$ and $\mathcal{G}$ are identical (Zhang et al., 2024, Proposition 2). Therefore, we have $Z_i \in Z_{\mathrm{mb}}$ if and only if $\hat{Z}_{\pi(i)} \in \hat{Z}_{\mathrm{mb}}$.

Now suppose $\hat{Z}_{\pi(i)} \in \hat{Z}_{\mathrm{mb}}$, which, by above reasoning, implies $Z_i \in Z_{\mathrm{mb}}$. By Statement (a), $\hat{Z}_{\pi(i)}$ is solely a function of a subset of $\{Z_i\} \cup \Psi_{Z_i}$. Here, we aim to show $\Psi_{Z_i} \subseteq Z_{\mathrm{mb}}$. Suppose $Z_j \in \Psi_{Z_i}$. By definition, $Z_i$ and $Y$ are adjacent in the Markov network $\mathcal{M}$, and thus $Z_j$ is also adjacent to $Y$ in $\mathcal{M}$ (because $Z_j$ is an intimate neighbor of $Z_i$). This implies $Z_j \in Z_{\mathrm{mb}}$. Therefore, we have $\{Z_i\} \cup \Psi_{Z_i} \subseteq Z_{\mathrm{mb}}$, i.e., $\hat{Z}_{\pi(i)}$ is solely a function of a subset of $Z_{\mathrm{mb}}$. Since this holds for every $\hat{Z}_{\pi(i)} \in \hat{Z}_{\mathrm{mb}}$, we conclude that $\hat{Z}_{\mathrm{mb}}$ is solely a function of a subset of $Z_{\mathrm{mb}}$.

Clearly, we can apply the same reasoning above (and Lemma 1) in the reverse direction to show that $Z_{\mathrm{mb}}$ is solely a function of a subset of $\hat{Z}_{\mathrm{mb}}$. Since the transformation from $Z$ to $\hat{Z}$ is a diffeomorphism, we conclude that $\hat{Z}_{\mathrm{mb}}$ is an invertible transformation of $Z_{\mathrm{mb}}$. $\square$

### A.1. Proof of Proposition 2

While the proof for the following proposition is inspired by Zhang et al. (2024, Proposition 1), ours involves a discrete target variable $Y$ (that is observed in the source domains), which requires the usage of Zheng et al. (2023, Theorem 2) to handle it.

**Proposition 2.** *Consider the generative process in Equation (1). Suppose that Assumptions 1 and 2 hold. Let $\hat{Z}$ and $\hat{\mathcal{M}}$ be the recovered latent variables and the recovered Markov network, respectively. By modeling the same generative process, we have the following statements:*

*(a) For each $Z_i$ and each $\{\hat{Z}_k, \hat{Z}_l\} \notin \mathcal{E}(\hat{\mathcal{M}})$, we have*

$$\frac{\partial Z_i}{\partial \hat{Z}_k} \frac{\partial Z_i}{\partial \hat{Z}_l} = 0.$$

*(b) For each $\{Z_i, Z_j\} \in \mathcal{E}(\mathcal{M})$ and each $\{\hat{Z}_k, \hat{Z}_l\} \notin \mathcal{E}(\hat{\mathcal{M}})$, we have*

$$\frac{\partial Z_i}{\partial \hat{Z}_k} \frac{\partial Z_j}{\partial \hat{Z}_l} = 0.$$

*(c) For each $\{Z_i, Y\} \in \mathcal{E}(\mathcal{M})$ and each $\{\hat{Z}_k, Y\} \notin \mathcal{E}(\hat{\mathcal{M}})$, we have*

$$\frac{\partial Z_i}{\partial \hat{Z}_k} = 0.$$

*Proof.* By definition, we have $X = g(Z)$ and $\hat{Z} = \hat{g}^{-1}(X)$, where $g$ and $\hat{g}$ are diffeomorphisms. Thus, the transformation from $Z$ to $\hat{Z}$, denoted by $v^{-1}$, is a diffeomorphism. Also, we have $\hat{Y} = Y$. By the change-of-variables formula, we obtain

$$\log P(\hat{Z}, \hat{Y}) = \log P(Z, Y) + \log |\det J_v|.$$

The first-order derivative is

$$\frac{\partial \log P(\hat{Z}, \hat{Y})}{\partial \hat{Z}_k} = \sum_{i=1}^{n} \frac{\partial \log P(Z, Y)}{\partial Z_i} \frac{\partial Z_i}{\partial \hat{Z}_k} + \frac{\partial \log |\det J_v|}{\partial \hat{Z}_k}. \tag{2}$$

Let $\hat{Z}_k$ and $\hat{Z}_l$ be latent variables that are not adjacent in the recovered Markov network $\hat{\mathcal{M}}$. The second-order derivative w.r.t. $\hat{Z}_k$ and $\hat{Z}_l$ is then given by

$$
\begin{aligned}
0 &= \sum_{j=1}^{n} \sum_{i=1}^{n} \frac{\partial^2 \log P(Z, Y)}{\partial Z_i \partial Z_j} \frac{\partial Z_j}{\partial \hat{Z}_l} \frac{\partial Z_i}{\partial \hat{Z}_k} + \sum_{i=1}^{n} \frac{\partial \log P(Z, Y)}{\partial Z_i} \frac{\partial^2 Z_i}{\partial \hat{Z}_k \partial \hat{Z}_l} + \frac{\partial^2 \log |\det J_v|}{\partial \hat{Z}_k \partial \hat{Z}_l} \\
&= \sum_{i=1}^{n} \frac{\partial^2 \log P(Z, Y)}{\partial Z_i^2} \frac{\partial Z_i}{\partial \hat{Z}_l} \frac{\partial Z_i}{\partial \hat{Z}_k} + \sum_{\substack{i,j: \\ i<j, \\ \{Z_i, Z_j\} \in \mathcal{E}(\mathcal{M})}} \frac{\partial^2 \log P(Z, Y)}{\partial Z_i \partial Z_j} \left( \frac{\partial Z_j}{\partial \hat{Z}_l} \frac{\partial Z_i}{\partial \hat{Z}_k} + \frac{\partial Z_i}{\partial \hat{Z}_l} \frac{\partial Z_j}{\partial \hat{Z}_k} \right) + \\
&\quad + \sum_{i=1}^{n} \frac{\partial \log P(Z, Y)}{\partial Z_i} \frac{\partial^2 Z_i}{\partial \hat{Z}_k \partial \hat{Z}_l} + \frac{\partial^2 \log |\det J_v|}{\partial \hat{Z}_k \partial \hat{Z}_l}.
\end{aligned}
$$

In the derivation above, we leveraged the following property (Lin, 1997): if $\hat{Z}_k$ and $\hat{Z}_l$ are not adjacent in the Markov network $\hat{\mathcal{M}}$, then they are conditionally independent given the remaining variables, which implies $\frac{\partial^2 \log P(\hat{Z}, \hat{Y})}{\partial \hat{Z}_k \partial \hat{Z}_l} = 0$. Similarly, this is also the case for $Z_i$ and $Z_j$.

Now consider the $u_r$ and $u_0$ domains where $r = 1, \ldots, 2n + |\mathcal{M}|$, and take the difference between the equations that correspond to them:

$$
\begin{aligned}
0 &= \sum_{i=1}^{n} \left( \frac{\partial^2 \log P^{(u_r)}(Z, Y)}{\partial Z_i^2} - \frac{\partial^2 \log P^{(u_0)}(Z, Y)}{\partial Z_i^2} \right) \frac{\partial Z_i}{\partial \hat{Z}_l} \frac{\partial Z_i}{\partial \hat{Z}_k} \\
&\quad + \sum_{\substack{i,j: \\ i<j, \\ \{Z_i, Z_j\} \in \mathcal{E}(\mathcal{M})}} \left( \frac{\partial^2 \log P^{(u_r)}(Z, Y)}{\partial Z_i \partial Z_j} - \frac{\partial^2 \log P^{(u_0)}(Z, Y)}{\partial Z_i \partial Z_j} \right) \left( \frac{\partial Z_j}{\partial \hat{Z}_l} \frac{\partial Z_i}{\partial \hat{Z}_k} + \frac{\partial Z_i}{\partial \hat{Z}_l} \frac{\partial Z_j}{\partial \hat{Z}_k} \right) + \\
&\quad + \sum_{i=1}^{n} \left( \frac{\partial \log P^{(u_r)}(Z, Y)}{\partial Z_i} - \frac{\partial \log P^{(u_0)}(Z, Y)}{\partial Z_i} \right) \frac{\partial^2 Z_i}{\partial \hat{Z}_k \partial \hat{Z}_l}.
\end{aligned}
$$

We collect the coefficients of the partial derivative terms in the equation above to form a vector, and consider the vectors for $r = 1, \ldots, 2n + |\mathcal{M}|$. Assumption A2 implies that these $2n + |\mathcal{M}|$ vectors are linearly independent. Therefore, for any $\{Z_i, Z_j\} \in \mathcal{E}(\mathcal{M})$ and $\{\hat{Z}_k, \hat{Z}_l\} \notin \mathcal{E}(\hat{\mathcal{M}})$, the following equations hold:

$$\frac{\partial Z_i}{\partial \hat{Z}_l} \frac{\partial Z_i}{\partial \hat{Z}_k} = 0, \tag{3}$$

$$\frac{\partial Z_j}{\partial \hat{Z}_l} \frac{\partial Z_i}{\partial \hat{Z}_k} + \frac{\partial Z_i}{\partial \hat{Z}_l} \frac{\partial Z_j}{\partial \hat{Z}_k} = 0, \tag{4}$$

$$\frac{\partial^2 Z_i}{\partial \hat{Z}_k \partial \hat{Z}_l} = 0.$$

Equation (3) implies that Statement (a) holds. By way of contradiction for Statement (b), suppose

$$\frac{\partial Z_j}{\partial \hat{Z}_l} \frac{\partial Z_i}{\partial \hat{Z}_k} \neq 0 \quad \implies \quad \frac{\partial Z_i}{\partial \hat{Z}_k} \neq 0, \tag{5}$$

which, with Equation (3), implies $\frac{\partial Z_i}{\partial \hat{Z}_l} = 0$. Substituting it into Equation (4), we have $\frac{\partial Z_j}{\partial \hat{Z}_l} \frac{\partial Z_i}{\partial \hat{Z}_k} = 0$, which is contradictory with Equation (5). Therefore, Equation (5) must not hold, i.e.,

$$\frac{\partial Z_j}{\partial \hat{Z}_l} \frac{\partial Z_i}{\partial \hat{Z}_k} = 0,$$

indicating that Statement (b) holds. It then remains to prove Statement (c).

Now suppose that $\hat{Z}_k$ and $\hat{Y}$ are not adjacent in the Markov network $\hat{\mathcal{M}}$. By Zheng et al. (2023, Theorem 2), for each $c_r \neq c_1$, we have

$$\frac{\partial \log P(\hat{Z}, \hat{Y} = v_{c_r})}{\partial \hat{Z}_k} - \frac{\partial \log P(\hat{Z}, \hat{Y} = v_{c_1})}{\partial \hat{Z}_k} = 0.$$

With Equation (2), we obtain

$$\begin{aligned}
0 &= \sum_{i=1}^n \left( \frac{\partial \log P^{(u)}(Z, Y = v_{c_r})}{\partial Z_i} - \frac{\partial \log P^{(u)}(Z, Y = v_{c_1})}{\partial Z_i} \right) \frac{\partial Z_i}{\partial \hat{Z}_k} \\
&= \sum_{i:\{Z_i, Y\} \in \mathcal{E}(\mathcal{M})} \left( \frac{\partial \log P^{(u)}(Z, Y = v_{c_r})}{\partial Z_i} - \frac{\partial \log P^{(u)}(Z, Y = v_{c_1})}{\partial Z_i} \right) \frac{\partial Z_i}{\partial \hat{Z}_k},
\end{aligned}$$

where the second line of the equation follows from the same property in Zheng et al. (2023, Theorem 2). Under Assumption 2, there exist $|Z_{\mathrm{mb}}|$ such equations above, and the $|Z_{\mathrm{mb}}|$ vectors formed by collecting those coefficients are linearly independent. This implies that Statement (c) holds, i.e.,

$$\frac{\partial Z_i}{\partial \hat{Z}_k} = 0.$$

$\square$

### A.2. Proof of Proposition 3

The proof for the following proposition is similar to Zhang et al. (2024, Theorem 2).

**Proposition 3** (Identifiability of Markov network). *Consider the generative process in Equation (1). Suppose that Assumptions 1 and 2 hold. By modeling the same generative process, the Markov network $\mathcal{M}$ is identifiable up to isomorphism.*

*Proof.* Since the transformation from $\hat{Z}$ to $Z$ is a diffeomorphism, there exists a permutation such that the diagonal entries in the permuted Jacobian matrix of such transformation are nonzero (e.g., see Zhang et al. (2024, Lemma 2) or Strang (2006; 2016)), which indicates

$$\frac{\partial Z_i}{\partial \hat{Z}_{\pi(i)}} \neq 0, \quad i = 1, \ldots, n. \tag{6}$$

Let $Z_i$ and $Z_j$ be two latent variables that are adjacent in the true Markov network $\mathcal{M}$, but $\hat{Z}_{\pi(i)}$ and $\hat{Z}_{\pi(j)}$ are not adjacent in the recovered Markov network $\hat{\mathcal{M}}$. With Proposition 2, we obtain

$$\frac{\partial Z_i}{\partial \hat{Z}_{\pi(i)}} \frac{\partial Z_j}{\partial \hat{Z}_{\pi(j)}} = 0,$$

which is contradictory with Equation (6). Now suppose $Z_i$ and $\hat{Y}$ are adjacent in the true Markov network $\mathcal{M}$, but $\hat{Z}_{\pi(i)}$ and $Y$ are not adjacent in the recovered Markov network $\hat{\mathcal{M}}$. With Proposition 2, we obtain

$$\frac{\partial Z_i}{\partial \hat{Z}_{\pi(i)}} = 0,$$

which is contradictory with Equation (6). Thus, we have proved that $\hat{\mathcal{M}}_\pi$ is a super-graph of $\mathcal{M}$, i.e., all edges in $\mathcal{M}$ are present in $\hat{\mathcal{M}}_\pi$. Since we apply sparsity constraint on $\hat{\mathcal{M}}$ during estimation such that it has smallest number of edges, we conclude that $\hat{\mathcal{M}}$ and $\mathcal{M}$ must be isomorphic. $\square$

### A.3. Proof of Proposition 4

**Proposition 4.** *Consider the generative process in Equation* (1). *Suppose that Assumptions 1 and 2 hold. Let $\hat{Z}$ and $\hat{\mathcal{M}}$ be the recovered latent variables and the recovered Markov network, respectively. By modeling the same generative process, we have the following statements:*

(a) *For each $Z_i$ and each $\{\hat{Z}_k, \hat{Z}_l\} \notin \mathcal{E}(\hat{\mathcal{M}})$, $Z_i$ is a function of at most one of $\hat{Z}_k$ and $\hat{Z}_l$.*

(b) *For each $\{Z_i, Z_j\} \in \mathcal{E}(\mathcal{M})$ and each $\{\hat{Z}_k, \hat{Z}_l\} \notin \mathcal{E}(\hat{\mathcal{M}})$, at most one of $Z_i$ and $Z_j$ is a function of $\hat{Z}_k$ and $\hat{Z}_l$.*

(c) *For each $\{Z_i, Y\} \in \mathcal{E}(\mathcal{M})$ and each $\{\hat{Z}_k, Y\} \notin \mathcal{E}(\hat{\mathcal{M}})$, $Z_i$ is not a function of $\hat{Z}_k$.*

*Sketch of proof.* By Proposition 2, for each $\{Z_i, Y\} \in \mathcal{E}(\mathcal{M})$ and each $\{\hat{Z}_k, Y\} \notin \mathcal{E}(\hat{\mathcal{M}})$, we have

$$\frac{\partial Z_i}{\partial \hat{Z}_k} = 0,$$

which implies that Statement (c) holds. Furthermore, by Statements (a) and (b) of Proposition 2, as well as Proposition 3, the same proof strategy of Zhang et al. (2024, Theorem 1) involving Intermediate Value Theorem can be used to show that Statements (a) and (b) of this proposition hold. $\square$

### A.4. Proof of Proposition 5

The proof here is partly inspired by Zhang et al. (2024, Theorem 3), while ours involves a discrete target variable $Y$ (that is observed in the source domains).

**Proposition 5** (Identifiability of latent variables). *Consider the generative process in Equation* (1). *Suppose that Assumptions 1 and 2 hold. Let $\hat{Z}$ be the recovered latent variables, and $\Psi_{Z_i}$ be the intimate neighbors of $Z_i$. By modeling the same generative process, there exists a permutation $\pi$ of $\hat{Z}$, denoted as $\hat{Z}_\pi$, such that $\hat{Z}_{\pi(i)}$ is solely a function of a subset of $\{Z_i\} \cup \Psi_{Z_i}$.*

*Proof.* We first prove the following lemma.

**Lemma 1.** *There exists a permutation $\pi$ of $\hat{Z}$, denoted as $\hat{Z}_\pi$, such that $Z_i$ is solely a function of a subset of $\hat{Z}_{\pi(i)} \cup \{\hat{Z}_{\pi(r)} \mid Z_r \in \Psi_{Z_i}\}$.*

Using Proposition 3 and its proof, there exists a permutation $\pi$ of $\hat{Z}$, denoted as $\hat{Z}_\pi$, such that the Markov networks $\mathcal{M}$ and $\hat{\mathcal{M}}_\pi$ are identical, and that $Z_i$ is a function of $\hat{Z}_{\pi(i)}$.

Suppose $Z_j$ is not adjacent to $Z_i$ in Markov network $\mathcal{M}$. This implies that $\hat{Z}_{\pi(i)}$ and $\hat{Z}_{\pi(j)}$ are not adjacent in $\hat{\mathcal{M}}$. Using Proposition 4, $Z_i$ is a function of at most one of $\hat{Z}_{\pi(i)}$ and $\hat{Z}_{\pi(j)}$. Since $Z_i$ is a function of $\hat{Z}_{\pi(i)}$ by definition, $Z_i$ must not be a function of $\hat{Z}_{\pi(j)}$.

Now suppose that $Z_j$ is adjacent to $Z_i$, but not adjacent to some other neighbor of $Z_i$. We consider the following two cases:

- Case 1: $Z_j$ is not adjacent to $Z_k$, while $Z_k$ is adjacent to $Z_i$. This implies that $\hat{Z}_{\pi(j)}$ and $\hat{Z}_{\pi(k)}$ are not adjacent in $\hat{\mathcal{M}}$. Using Proposition 4, at most one of $Z_i$ and $Z_k$ is a function of $\hat{Z}_{\pi(j)}$ and $\hat{Z}_{\pi(k)}$. Since $Z_k$ is a function of $\hat{Z}_{\pi(k)}$ by definition, $Z_i$ cannot be a function of $\hat{Z}_{\pi(j)}$.

- Case 2: $Z_j$ is not adjacent to $Y$, while $Y$ is adjacent to $Z_i$. This implies that $\hat{Z}_{\pi(j)}$ and $Y$ are not adjacent in $\hat{\mathcal{M}}$. Using Proposition 4, $Z_i$ cannot be a function of $\hat{Z}_{\pi(j)}$.

Thus, we have proved Lemma 1. Suppose $Z_r \notin \{Z_i\} \cup \Psi_{Z_i}$, which, by Lemma 1, implies that $Z_i$ cannot be a function of $\hat{Z}_{\pi(r)}$, i.e.,

$$\left(\frac{\partial Z}{\partial \hat{Z}_\pi}\right)_{ir} = \frac{\partial Z_i}{\partial \hat{Z}_{\pi(r)}} = 0.$$

Using Zhang et al. (2024, Proposition 3) w.r.t. $\frac{\partial Z}{\partial \hat{Z}_\pi}$, we conclude that

$$\left(\frac{\partial Z}{\partial \hat{Z}_\pi}\right)_{ir}^{-1} = 0$$

and therefore

$$\frac{\partial \hat{Z}_{\pi(i)}}{\partial Z_r} = \left(\frac{\partial \hat{Z}_\pi}{\partial Z}\right)_{ir} = \left(\frac{\partial Z}{\partial \hat{Z}_\pi}\right)_{ir}^{-1} = 0.$$

That is, $\hat{Z}_{\pi(i)}$ must not be a function of $Z_r$. This implies that $\hat{Z}_{\pi(i)}$ is solely a function of a subset of $\{Z_i\} \cup \Psi_{Z_i}$. $\qquad\square$

## B. Proof of Theorem 2

The proof of the following theorem shares similar spirit with that of Theorem 1.

**Theorem 2** (Subspace identifiability of parents, children, and spouses). *Consider the generative process in Equation* (1). *Suppose that Assumptions 1, 2 and 3, as well as the faithfulness assumption, hold. By modeling the same generative process with minimal number of edges for the learned Markov network $\hat{\mathcal{M}}$, there exists a partition of the learned Markov blanket $\hat{Z}_{\mathrm{mb}}$, denoted as $\hat{Z}_{S_1}$, $\hat{Z}_{S_2}$, and $\hat{Z}_{S_3}$, such that they are invertible transformations of the true parents $Z_{\mathrm{pa}}$, children $Z_{\mathrm{ch}}$, and spouses $Z_{\mathrm{sps}}$, respectively.*

*Proof.* Recall that $\hat{Z}$ denotes the recovered latent variables, $\hat{\mathcal{M}}$ denotes the recovered Markov network, and $\Psi_{Z_i}$ denotes the intimate neighbors of $Z_i$. By Propositions 3 and 5, there exists a permutation $\pi$ of $\hat{Z}$, denoted as $\hat{Z}_\pi$, such that the following statements hold:

(a) $\hat{Z}_{\pi(i)}$ is solely a function of a subset of $\{Z_i\} \cup \Psi_{Z_i}$.

(b) $\hat{\mathcal{M}}_\pi$ and $\mathcal{M}$ are identical.

By Statement (b), under the faithfulness assumption (specifically the SAF and SUCF assumptions), the moralized graphs of $\hat{\mathcal{G}}$ and $\mathcal{G}$ are identical (Zhang et al., 2024, Proposition 2). Therefore, we have $Z_i \in Z_{\mathrm{mb}}$ if and only if $\hat{Z}_{\pi(i)} \in \hat{Z}_{\mathrm{mb}}$.

Consider a partition of $\hat{Z}_{\mathrm{mb}}$, denoted as $\hat{Z}_{S_1}$, $\hat{Z}_{S_2}$, and $\hat{Z}_{S_3}$, where

$$\hat{Z}_{S_1} := \{\hat{Z}_{\pi(k)} \mid Z_k \in Z_{\mathrm{pa}}\}, \qquad \hat{Z}_{S_2} := \{\hat{Z}_{\pi(k)} \mid Z_k \in Z_{\mathrm{ch}}\}, \qquad \text{and} \qquad \hat{Z}_{S_3} := \{\hat{Z}_{\pi(k)} \mid Z_k \in Z_{\mathrm{sps}}\}.$$

Now suppose $\hat{Z}_{\pi(i)} \in \hat{Z}_{S_1}$, which, by definition, implies $Z_i \in Z_{\mathrm{pa}}$. By Statement (a), $\hat{Z}_{\pi(i)}$ is solely a function of a subset of $\{Z_i\} \cup \Psi_{Z_i}$. Under Assumption 3, we have $\Psi_{Z_i} \subseteq Z_{\mathrm{pa}}$. This implies $\{Z_i\} \cup \Psi_{Z_i} \subseteq Z_{\mathrm{pa}}$, i.e., $\hat{Z}_{\pi(i)}$ is solely a function of a subset of $Z_{\mathrm{pa}}$. Since this holds for every $\hat{Z}_{\pi(i)} \in \hat{Z}_{S_1}$, we conclude that $\hat{Z}_{S_1}$ is solely a function of a subset of $Z_{\mathrm{pa}}$. Clearly, we can apply the same reasoning (and Lemma 1) in the reverse direction to show that $Z_{\mathrm{pa}}$ is solely a function of a subset of $\hat{Z}_{S_1}$. Since the transformation from $Z$ to $\hat{Z}$ is a diffeomorphism, we conclude that $\hat{Z}_{S_1}$ is an invertible transformation of $Z_{\mathrm{pa}}$.

The same reasoning above can be used to show that $\hat{Z}_{S_2}$ and $\hat{Z}_{S_3}$ are invertible transformations of $Z_{\mathrm{ch}}$ and $Z_{\mathrm{sps}}$, respectively. $\qquad\square$

## C. Proof of Proposition 1 and Corollary 1

### C.1. Proof of Proposition 1

**Proposition 1.** *Consider the generative process in Equation* (1). *We have*

$$P^\tau(Y = v_k \mid X) = \frac{P^\tau(Z_{\mathrm{ch}} \mid Y = v_k, Z_{\mathrm{sps}})P^\tau(Y = v_k \mid Z_{\mathrm{pa}})}{\sum_{c=1}^{C} P^\tau(Z_{\mathrm{ch}} \mid Y = v_c, Z_{\mathrm{sps}})P^\tau(Y = v_c \mid Z_{\mathrm{pa}})}.$$

*Proof.* We have

$$
\begin{aligned}
P^\tau(Y = v_k \mid X) &= \frac{P^\tau(Y = v_k, X)}{P^\tau(X)} \\
&= \frac{P^\tau(Y = v_k, Z)}{P^\tau(Z)} && \text{(Change-of-variables)} \\
&= P^\tau(Y = v_k \mid Z) \\
&= P^\tau(Y = v_k \mid Z_{\mathrm{mb}}, Z_{\mathrm{mb}}^{\complement}) \\
&= P^\tau(Y = v_k \mid Z_{\mathrm{mb}}) && (\because Y \perp\!\!\!\perp Z_{\mathrm{mb}}^{\complement} \mid Z_{\mathrm{mb}}) \\
&= \frac{P^\tau(Y = v_k, Z_{\mathrm{mb}})}{P^\tau(Z_{\mathrm{mb}})} \\
&= \frac{P^\tau(Y = v_k, Z_{\mathrm{mb}})}{\sum_{c=1}^{C} P^\tau(Y = v_c, Z_{\mathrm{mb}})} \\
&= \frac{P^\tau(Y = v_k, Z_{\mathrm{pa}}, Z_{\mathrm{sps}}, Z_{\mathrm{ch}})}{\sum_{c=1}^{C} P^\tau(Y = v_c, Z_{\mathrm{pa}}, Z_{\mathrm{sps}}, Z_{\mathrm{ch}})} \\
&= \frac{P^\tau(Z_{\mathrm{ch}} \mid Y = v_k, Z_{\mathrm{pa}}, Z_{\mathrm{sps}}) P^\tau(Y = v_k \mid Z_{\mathrm{pa}}, Z_{\mathrm{sps}}) P^\tau(Z_{\mathrm{pa}}, Z_{\mathrm{sps}})}{\sum_{c=1}^{C} P^\tau(Z_{\mathrm{ch}} \mid Y = v_c, Z_{\mathrm{pa}}, Z_{\mathrm{sps}}) P^\tau(Y = v_c \mid Z_{\mathrm{pa}}, Z_{\mathrm{sps}}) P^\tau(Z_{\mathrm{pa}}, Z_{\mathrm{sps}})} \\
&= \frac{P^\tau(Z_{\mathrm{ch}} \mid Y = v_k, Z_{\mathrm{sps}}) P^\tau(Y = v_k \mid Z_{\mathrm{pa}})}{\sum_{c=1}^{C} P^\tau(Z_{\mathrm{ch}} \mid Y = v_c, Z_{\mathrm{sps}}) P^\tau(Y = v_c \mid Z_{\mathrm{pa}})}.
\end{aligned}
$$

In the last step, we use the conditional independence relations $Z_{\mathrm{ch}} \perp\!\!\!\perp Z_{\mathrm{pa}} \mid Y, Z_{\mathrm{sps}}$ and $Y \perp\!\!\!\perp Z_{\mathrm{sps}} \mid Z_{\mathrm{pa}}$. $\qquad\square$

### C.2. Proof of Corollary 1

**Corollary 1.** *Consider the generative process in Equation* (1)*. Let $\hat{Z}_{\mathrm{pa}}$, $\hat{Z}_{\mathrm{ch}}$, and $\hat{Z}_{\mathrm{sps}}$ be invertible transformations of $Z_{\mathrm{pa}}$, $Z_{\mathrm{ch}}$, and $Z_{\mathrm{sps}}$, respectively. We have*

$$
P^\tau(Y = v_k \mid X) = \frac{P^\tau(\hat{Z}_{\mathrm{ch}} \mid Y = v_k, \hat{Z}_{\mathrm{sps}}) P^\tau(Y = v_k \mid \hat{Z}_{\mathrm{pa}})}{\sum_{c=1}^{C} P^\tau(\hat{Z}_{\mathrm{ch}} \mid Y = v_c, \hat{Z}_{\mathrm{sps}}) P^\tau(Y = v_c \mid \hat{Z}_{\mathrm{pa}})}.
$$

*Proof.* By Proposition 1 and the change-of-variables formula, we have

$$
\begin{aligned}
P^\tau(Y = v_k \mid X) &= \frac{P^\tau(Z_{\mathrm{ch}} \mid Y = v_k, Z_{\mathrm{sps}}) P^\tau(Y = v_k \mid Z_{\mathrm{pa}})}{\sum_{c=1}^{C} P^\tau(Z_{\mathrm{ch}} \mid Y = v_c, Z_{\mathrm{sps}}) P^\tau(Y = v_c \mid Z_{\mathrm{pa}})} \\
&= \frac{\frac{P^\tau(Z_{\mathrm{ch}}, Y = v_k, Z_{\mathrm{sps}})}{P^\tau(Y = v_k, Z_{\mathrm{sps}})} \frac{P^\tau(Y = v_k, Z_{\mathrm{pa}})}{P^\tau(Z_{\mathrm{pa}})}}{\sum_{c=1}^{C} \frac{P^\tau(Z_{\mathrm{ch}}, Y = v_c, Z_{\mathrm{sps}})}{P^\tau(Y = v_c, Z_{\mathrm{sps}})} \frac{P^\tau(Y = v_c, Z_{\mathrm{pa}})}{P^\tau(Z_{\mathrm{pa}})}} \\
&= \frac{\frac{P^\tau(\hat{Z}_{\mathrm{ch}}, Y = v_k, \hat{Z}_{\mathrm{sps}})}{P^\tau(Y = v_k, \hat{Z}_{\mathrm{sps}})} \frac{P^\tau(Y = v_k, \hat{Z}_{\mathrm{pa}})}{P^\tau(\hat{Z}_{\mathrm{pa}})}}{\sum_{c=1}^{C} \frac{P^\tau(\hat{Z}_{\mathrm{ch}}, Y = v_c, \hat{Z}_{\mathrm{sps}})}{P^\tau(Y = v_c, \hat{Z}_{\mathrm{sps}})} \frac{P^\tau(Y = v_c, \hat{Z}_{\mathrm{pa}})}{P^\tau(\hat{Z}_{\mathrm{pa}})}} \\
&= \frac{P^\tau(\hat{Z}_{\mathrm{ch}} \mid Y = v_k, \hat{Z}_{\mathrm{sps}}) P^\tau(Y = v_k \mid \hat{Z}_{\mathrm{pa}})}{\sum_{c=1}^{C} P^\tau(\hat{Z}_{\mathrm{ch}} \mid Y = v_c, \hat{Z}_{\mathrm{sps}}) P^\tau(Y = v_c \mid \hat{Z}_{\mathrm{pa}})}.
\end{aligned}
$$

$\qquad\square$

## D. Proof of Theorem 3

The proof of the following theorem is partly inspired by Stojanov et al. (2019).

**Theorem 3** (Identifiability of target distribution)**.** *Suppose that Assumptions 4 and 5 hold. Let $\hat{Z}_{\mathrm{pa}}$, $\hat{Z}_{\mathrm{ch}}$, and $\hat{Z}_{\mathrm{sps}}$ be invertible transformations of $Z_{\mathrm{pa}}$, $Z_{\mathrm{ch}}$, and $Z_{\mathrm{sps}}$, respectively. Suppose that we learn $P^{new}$ to match $P^\tau(\hat{Z}_{\mathrm{ch}} \mid \hat{Z}_{\mathrm{pa}}, \hat{Z}_{\mathrm{sps}})$*

in the target domain, i.e., $P^{new}(\hat{Z}_{\text{ch}} \mid \hat{Z}_{\text{pa}}, \hat{Z}_{\text{sps}}) = P^{\tau}(\hat{Z}_{\text{ch}} \mid \hat{Z}_{\text{pa}}, \hat{Z}_{\text{sps}})$ while constraining $P^{new}(\hat{Z}_{\text{ch}} \mid Y, \hat{Z}_{\text{sps}})$ to satisfy Assumption 4. Then, we have $P^{\tau}(\hat{Z}_{\text{ch}} \mid Y, \hat{Z}_{\text{sps}}) = P^{new}(\hat{Z}_{\text{ch}} \mid Y, \hat{Z}_{\text{sps}})$ and $P^{\tau}(Y \mid \hat{Z}_{\text{pa}}) = P^{new}(Y \mid \hat{Z}_{\text{pa}})$.

*Proof.* We first have

$$P^{\tau}(\hat{Z}_{\text{ch}} \mid \hat{Z}_{\text{pa}}, \hat{Z}_{\text{sps}}) = P^{\text{new}}(\hat{Z}_{\text{ch}} \mid \hat{Z}_{\text{pa}}, \hat{Z}_{\text{sps}})$$

$$\frac{P^{\tau}(\hat{Z}_{\text{ch}}, \hat{Z}_{\text{pa}}, \hat{Z}_{\text{sps}})}{P^{\tau}(\hat{Z}_{\text{pa}}, \hat{Z}_{\text{sps}})} = \frac{P^{\text{new}}(\hat{Z}_{\text{ch}}, \hat{Z}_{\text{pa}}, \hat{Z}_{\text{sps}})}{P^{\text{new}}(\hat{Z}_{\text{pa}}, \hat{Z}_{\text{sps}})}.$$

By the change-of-variables formula and further simplifying, we have

$$\frac{P^{\tau}(Z_{\text{ch}}, Z_{\text{pa}}, Z_{\text{sps}})}{P^{\tau}(Z_{\text{pa}}, Z_{\text{sps}})} = \frac{P^{\text{new}}(Z_{\text{ch}}, Z_{\text{pa}}, Z_{\text{sps}})}{P^{\text{new}}(Z_{\text{pa}}, Z_{\text{sps}})}$$

$$\frac{\sum_{c=1}^{C} P^{\tau}(Z_{\text{ch}}, Z_{\text{pa}}, Z_{\text{sps}}, Y = v_c)}{P^{\tau}(Z_{\text{pa}}, Z_{\text{sps}})} = \frac{\sum_{c=1}^{C} P^{\text{new}}(Z_{\text{ch}}, Z_{\text{pa}}, Z_{\text{sps}}, Y = v_c)}{P^{\text{new}}(Z_{\text{pa}}, Z_{\text{sps}})}$$

$$\frac{\sum_{c=1}^{C} P^{\tau}(Z_{\text{ch}} \mid Y = v_c, Z_{\text{sps}}) P^{\tau}(Y = v_c \mid Z_{\text{pa}}) P^{\tau}(Z_{\text{pa}}, Z_{\text{sps}})}{P^{\tau}(Z_{\text{pa}}, Z_{\text{sps}})} = \frac{\sum_{c=1}^{C} P^{\text{new}}(Z_{\text{ch}} \mid Y = v_c, Z_{\text{sps}}) P^{\text{new}}(Y = v_c \mid Z_{\text{pa}})}{P^{\text{new}}(Z_{\text{pa}}, Z_{\text{sps}}) P^{\text{new}}(Z_{\text{pa}}, Z_{\text{sps}})}$$

$$\sum_{c=1}^{C} P^{\tau}(Y = v_c \mid Z_{\text{pa}}) P^{\tau}(Z_{\text{ch}} \mid Y = v_c, Z_{\text{sps}}) = \sum_{c=1}^{C} P^{\text{new}}(Y = v_c \mid Z_{\text{pa}}) P^{\text{new}}(Z_{\text{ch}} \mid Y = v_c, Z_{\text{sps}}).$$

Applying Assumption 4 for $P^{\tau}(Z_{\text{ch}} \mid Y = v_c, Z_{\text{sps}})$ and $P^{\text{new}}(Z_{\text{ch}} \mid Y = v_c, Z_{\text{sps}})$, we obtain

$$\sum_{c=1}^{C} P^{\tau}(Y = v_c \mid Z_{\text{pa}}) P^{\alpha_c^{\tau}}(Z_{\text{ch}} \mid Y = v_c, Z_{\text{sps}}) = \sum_{c=1}^{C} P^{\text{new}}(Y = v_c \mid Z_{\text{pa}}) P^{\alpha_c^{\text{new}}}(Z_{\text{ch}} \mid Y = v_c, Z_{\text{sps}}),$$

which implies

$$\sum_{c=1}^{C} \left( P^{\tau}(Y = v_c \mid Z_{\text{pa}}) P^{\alpha_c^{\tau}}(Z_{\text{ch}} \mid Y = v_c, Z_{\text{sps}}) - P^{\text{new}}(Y = v_c \mid Z_{\text{pa}}) P^{\alpha_c^{\text{new}}}(Z_{\text{ch}} \mid Y = v_c, Z_{\text{sps}}) \right) = 0.$$

By Assumption 5, we have

$$P^{\tau}(Y = v_c \mid Z_{\text{pa}}) P^{\alpha_c^{\tau}}(Z_{\text{ch}} \mid Y = v_c, Z_{\text{sps}}) - P^{\text{new}}(Y = v_c \mid Z_{\text{pa}}) P^{\alpha_c^{\text{new}}}(Z_{\text{ch}} \mid Y = v_c, Z_{\text{sps}}) = 0, \tag{7}$$

which, by taking integral w.r.t. $Z_{\text{ch}}$, indicates

$$P^{\tau}(Y = v_c \mid Z_{\text{pa}}) = P^{\text{new}}(Y = v_c \mid Z_{\text{pa}}). \tag{8}$$

Plugging the above equation into Equation (7) yields

$$P^{\alpha_c^{\tau}}(Z_{\text{ch}} \mid Y = v_c, Z_{\text{sps}}) = P^{\alpha_c^{\text{new}}}(Z_{\text{ch}} \mid Y = v_c, Z_{\text{sps}}),$$

or, equivalently,

$$P^{\text{new}}(Z_{\text{ch}} \mid Y = v_c, Z_{\text{sps}}) = P^{\tau}(Z_{\text{ch}} \mid Y = v_c, Z_{\text{sps}}). \tag{9}$$

Applying change-of-variables formula to Equations (9) and (8), we obtain

$$P^{\tau}(\hat{Z}_{\text{ch}} \mid Y, \hat{Z}_{\text{sps}}) = P^{\text{new}}(\hat{Z}_{\text{ch}} \mid Y, \hat{Z}_{\text{sps}}) \quad \text{and} \quad P^{\tau}(Y \mid \hat{Z}_{\text{pa}}) = P^{\text{new}}(Y \mid \hat{Z}_{\text{pa}}).$$

$\square$

# E. Experimental Details, Analysis and More Experiments

**Model details.**    Our proposed approach adopts a hierarchical VAE architecture with the following detailed module designs. The domain size is $M$ and the number of categories is $C$. The **primary VAE encoder** consists of a fully connected layer (backbone features $\rightarrow$ hidden dimension) with batch normalization and ReLU activation, followed by two linear projections to generate mean $\mu$ and log-variance $\log \sigma^2$ for the latent space $Z \in \mathbb{R}^{d_o + d_{Z_{\mathrm{pa}}} + d_{Z_{\mathrm{ch}}} + d_{Z_{\mathrm{sps}}}}$, where $d_o$, $d_{Z_{\mathrm{pa}}}$, $d_{Z_{\mathrm{ch}}}$, and $d_{Z_{\mathrm{sps}}}$ denote the dimensions we set for $Z_{\mathrm{mb}}^{\mathsf{C}}$, $Z_{\mathrm{pa}}$, $Z_{\mathrm{ch}}$, and $Z_{\mathrm{sps}}$, respectively. The **decoder** reconstructs features through a two-layer MLP (latent dimension $\rightarrow$ hidden dimension $\rightarrow$ backbone feature dimension) with batch normalization and ReLU. **Domain-specific embeddings** are implemented as learnable embedding layers: $\boldsymbol{\theta}_Y \in \mathbb{R}^{M \times d_{\theta_Y}}$ and $\boldsymbol{\theta}_{\mathrm{ch}} \in \mathbb{R}^{M \times d_{\theta_{\mathrm{ch}}}}$, where $d_{\theta_Y}$ and $d_{\theta_{\mathrm{ch}}}$ denote the dimensions we set for $\theta_Y$ and $\theta_{\mathrm{ch}}$, respectively. The **auxiliary VAE modules** use single linear layers in both the encoder and decoder, which operate on the concatenated vector $(\theta_Y, Z_{\mathrm{pa}}) \in \mathbb{R}^{d_{\theta_Y} + d_{Z_{\mathrm{pa}}}}$ to predict/reconstruct class distributions. Similarly, we use encoder and decoder with linear layers to handle $(\theta_{\mathrm{ch}}, Y, Z_{\mathrm{sps}}) \in \mathbb{R}^{d_{\theta_{\mathrm{ch}}} + C + d_{Z_{\mathrm{sps}}}}$ for $Z_{ch}$ reconstruction. The **final classifier** is a two-layer MLP that processes concatenated features $(Z_{\mathrm{pa}}, Z_{\mathrm{ch}}, Z_{\mathrm{sps}}, \theta_Y, \theta_{\mathrm{ch}})$ through a hidden layer with ReLU activation to output class predictions. All backbone features undergo adaptive average pooling and flattening before processing.

**Computing resources and efficiency.**    We train our model using a NVIDIA A100-SXM4-40GB GPU. For the Office-Home dataset, the batch size is set to 32, and the model is trained for 70 epochs, which takes approximately 160 minutes. The peak memory usage is around 35 GB. The majority of the computational cost comes from the ResNet-50 backbone, as we only add several lightweight MLP layers after it. For the PACS dataset, the batch size is set to 32, and the model is trained for 70 epochs, each epoch has 200 steps, which takes approximately 32 minutes. The peak memory usage is around 11 GB.

**Visualization and standard deviation.**    We have conducted visualizations of the latent space of features and VAE. Specifically, the t-SNE visualizations of the learned features on the Clipart task from the Office-Home dataset are available in Figure 3, which demonstrate the effectiveness of our method at aligning the source and target domains while preserving discriminative structures. We also report the standard deviations for Office–Home and PACS datasets in Tables 3 and 4, respectively. In particular, GAMA not only achieves the highest average accuracy but also exhibits very low variance, demonstrating its stable performance across different subtasks.

Table 3: Office–Home dataset results (accuracy $\pm$ std).

| Method | Ar | Cl | Pr | Rw | Avg |
|---|---|---|---|---|---|
| DAN (Long et al., 2015) | $68.3 \pm 0.5$ | $57.9 \pm 0.7$ | $78.5 \pm 0.1$ | $81.9 \pm 0.4$ | 71.6 |
| Source Only (He et al., 2016) | $64.6 \pm 0.7$ | $52.3 \pm 0.6$ | $77.6 \pm 0.2$ | $80.7 \pm 0.8$ | 68.8 |
| DANN (Ganin et al., 2016) | $64.3 \pm 0.6$ | $58.0 \pm 1.6$ | $76.4 \pm 0.5$ | $78.8 \pm 0.5$ | 69.4 |
| DCTN (Xu et al., 2018) | $66.9 \pm 0.6$ | $61.8 \pm 0.5$ | $79.2 \pm 0.6$ | $77.8 \pm 0.6$ | 71.4 |
| MCD (Saito et al., 2018) | $67.8 \pm 0.4$ | $59.9 \pm 0.6$ | $79.2 \pm 0.6$ | $80.9 \pm 0.2$ | 72.0 |
| DANN+BSP (Chen et al., 2019c) | $66.1 \pm 0.3$ | $61.0 \pm 0.4$ | $78.1 \pm 0.3$ | $79.9 \pm 0.1$ | 71.3 |
| M3SDA (Peng et al., 2019) | $66.2 \pm 0.5$ | $58.6 \pm 0.6$ | $79.5 \pm 0.5$ | $81.4 \pm 0.2$ | 71.4 |
| iMSDA (Kong et al., 2022) | $75.4 \pm 0.9$ | $61.4 \pm 0.7$ | $83.5 \pm 0.2$ | $84.5 \pm 0.4$ | 76.2 |
| **GAMA (Ours)** | $76.6 \pm 0.1$ | $62.6 \pm 0.6$ | $84.9 \pm 0.1$ | $84.9 \pm 0.1$ | 77.3 |

Table 4: PACS dataset results (accuracy $\pm$ std).

| Method | Art | Cartoon | Photo | Sketch | Avg |
|---|---|---|---|---|---|
| Source Only (He et al., 2016) | $74.9 \pm 0.88$ | $72.1 \pm 0.75$ | $94.5 \pm 0.58$ | $64.7 \pm 1.53$ | 76.6 |
| DANN (Ganin et al., 2016) | $81.9 \pm 1.13$ | $77.5 \pm 1.26$ | $91.8 \pm 1.21$ | $74.6 \pm 1.03$ | 81.5 |
| MDAN (Zhao et al., 2018) | $79.1 \pm 0.36$ | $76.0 \pm 0.73$ | $91.4 \pm 0.85$ | $72.0 \pm 0.80$ | 79.6 |
| WBN (Mancini et al., 2018) | $89.9 \pm 0.28$ | $89.7 \pm 0.56$ | $97.4 \pm 0.84$ | $58.0 \pm 1.51$ | 83.8 |
| MCD (Saito et al., 2018) | $88.7 \pm 1.01$ | $88.9 \pm 1.53$ | $96.4 \pm 0.42$ | $73.9 \pm 3.94$ | 87.0 |
| M3SDA (Peng et al., 2019) | $89.3 \pm 0.42$ | $89.9 \pm 1.00$ | $97.3 \pm 0.31$ | $76.7 \pm 2.86$ | 88.3 |
| CMSS (Yang et al., 2020) | $88.6 \pm 0.36$ | $90.4 \pm 0.80$ | $96.9 \pm 0.27$ | $82.0 \pm 0.59$ | 89.5 |
| iMSDA (Kong et al., 2022) | $93.75 \pm 0.32$ | $92.46 \pm 0.23$ | $98.48 \pm 0.07$ | $89.22 \pm 0.73$ | 93.48 |
| **GAMA (Ours)** | $98.77 \pm 0.11$ | $93.73 \pm 0.75$ | $92.81 \pm 0.40$ | $89.27 \pm 0.68$ | **93.65** |

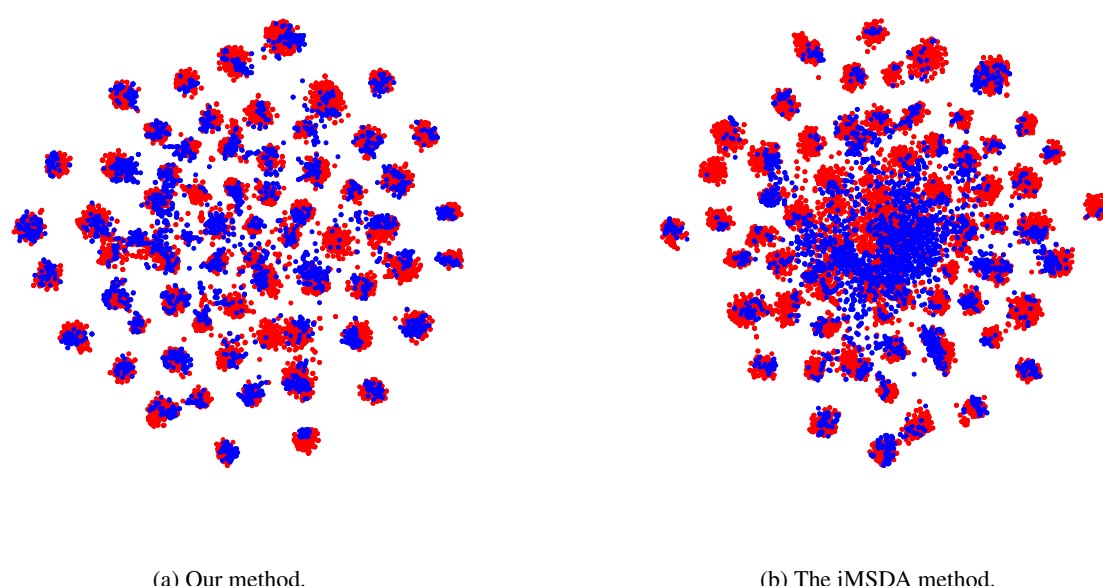

(a) Our method.

(b) The iMSDA method.

Figure 3: The t-SNE visualizations of the learned features on the $\rightarrow$Clipart task in the Office-Home dataset. Specifically, red points indicate learned features form the source domains, while blue points indicate learned features from the target domain.

Table 5: Hyperparameters for Office-Home (Ar, Cl, Pr, Rw) and PACS (P, A, C, S) datasets.

| Parameter | Office-Home | | | | PACS | | | |
|---|---|---|---|---|---|---|---|---|
| | Ar | Cl | Pr | Rw | P | A | C | S |
| $\lambda_1$ | $2 \times 10^{-3}$ | $6 \times 10^{-4}$ | $3 \times 10^{-3}$ | $6 \times 10^{-4}$ | $7 \times 10^{-4}$ | $3 \times 10^{-3}$ | $5 \times 10^{-3}$ | $9 \times 10^{-3}$ |
| $\lambda_2$ | $4 \times 10^{-4}$ | $2 \times 10^{-4}$ | $3 \times 10^{-3}$ | $1 \times 10^{-4}$ | $4 \times 10^{-4}$ | $1 \times 10^{-4}$ | $1 \times 10^{-4}$ | $1 \times 10^{-3}$ |
| $\lambda_3$ | $1 \times 10^{-4}$ | $8 \times 10^{-4}$ | $1 \times 10^{-3}$ | $4 \times 10^{-3}$ | $5 \times 10^{-4}$ | $2 \times 10^{-3}$ | $2 \times 10^{-4}$ | $7 \times 10^{-4}$ |
| $\lambda_4$ | $5 \times 10^{-3}$ | $6 \times 10^{-4}$ | $7 \times 10^{-3}$ | $1 \times 10^{-3}$ | $1 \times 10^{-3}$ | $2 \times 10^{-4}$ | $4 \times 10^{-4}$ | $5 \times 10^{-3}$ |
| $\lambda_5$ | $2 \times 10^{-3}$ | $4 \times 10^{-4}$ | $3 \times 10^{-4}$ | $4 \times 10^{-3}$ | $4 \times 10^{-3}$ | $9 \times 10^{-4}$ | $4 \times 10^{-3}$ | $5 \times 10^{-3}$ |

