# OpenReview forum: "A General Representation-Based Approach to Multi-Source Domain Adaptation"
_ICML.cc/2025/Conference — ICML 2025 poster_

### Official Review · Reviewer_d2cE · 2025-03-12

**Overall Recommendation:** 3

**Summary:**

This paper addresses the issue of multi-source domain adaptation with a focus on identifiability. It introduces a causal framework that avoids restrictive assumptions such as independent latent variables or invariant label distributions. The authors theoretically establish identifiability and validate the effectiveness of their method through experiments on datasets like Office-Home and PACS.

**Claims And Evidence:**

All the claims are supported by clear and convincing evidence.

**Essential References Not Discussed:**

As far as I known, all the essential references are discussed in the paper.

**Experimental Designs Or Analyses:**

1. The primary concern with this work is that while it claims to "consider the most general scenario" and suggests adaptability to different data distribution shifts, the experiments are limited to datasets with covariate shifts. It would be more convincing if the authors also conducted experiments involving other types of distribution shifts, such as label shift or conditional shift.
2. The most recent domain adaptation (DA) baselines included in the study are from 2022. Including the latest baselines would enhance the rigor of the analysis.

**Methods And Evaluation Criteria:**

The methods and evaluation criteria are suitable for the problem.

**Other Comments Or Suggestions:**

None

**Other Strengths And Weaknesses:**

**Strengths:**

1. The novel multi-source adaptation framework introduced in the paper guarantees identifiability under mild restrictions.
2. Effectiveness of the theorem and its implementation are verified through experiments conducted on datasets such as Office-Home and PACS.
3. The paper is well-written and easy to follow.

**Other Weaknesses:**

None. Most potential weaknesses are addressed in the "Experimental Designs" section.

**Questions For Authors:**

The theoretical framework proposed by the author suggests there are multiple ways to implement it in practice. Why did the authors choose a multi-VAE structure for implementation? Could the author provide more insights into their intuition behind this choice?

**Relation To Broader Scientific Literature:**

The works most closely related to this are [1] and [2]. However, [1] adopts more stringent assumptions about the independent latent variables, whereas [2] does not ensure identifiability.

[1] Kong, Lingjing, et al. "Partial disentanglement for domain adaptation." International conference on machine learning. PMLR, 2022.
[2] Zhang, Kun, et al. "Domain adaptation as a problem of inference on graphical models." Advances in neural information processing systems 33 (2020): 4965-4976.

**Theoretical Claims:**

Due to limited time, I couldn't check every detail of the derivation. However, most of the theorems seem intuitive and appear to be correct.

---

> ### Author Rebuttal · Authors · 2025-04-01
>
> We sincerely appreciate the reviewer's constructive comments, helpful feedback, and time devoted. Please see below for our response.
>
> **Q1:** "the experiments are limited to datasets with covariate shifts" and "experiments involving other types of distribution shifts"
>
> **A1:** Thanks for your comment. Notably, several existing works [1, 2, 3] on label shifts have also used Office-Home and PACS, highlighting the presence of label shifts in these datasets. A possible reason is that, to the best of our knowledge, there is no benchmark dataset specifically designed for label shift. If you are aware of any such datasets, we would greatly appreciate your suggestions and would be happy to incorporate them into our analysis.
>
> To validate the presence of label shifts, we analyzed the label distributions of Office-Home and PACS. The visualizations are provided in Figures 1, 2, and 3 in https://anonymous.4open.science/r/icml-rebuttal-gama/rebuttal.pdf. Specifically, Figures 1 and 2 illustrate the label distributions across different domains for Office-Home and PACS, respectively, indicating notable variations in label probabilities across domains. To further quantify this, we computed the Jensen–Shannon divergence between label distributions for different domain pairs (Figure 3), confirming clear label shifts—especially in the Art sub-task of Office-Home and the Sketch sub-task of PACS. Thus, beyond covariate shift as you pointed out, both datasets also exhibit label shifts.
>
> To further investigate our method’s robustness to severe label shift, we created a more extreme setting for Clipart sub-task (from Office-Home dataset ) by sampling data points from different labels according to a pre-defined label distribution; see Figure 4 in https://anonymous.4open.science/r/icml-rebuttal-gama/rebuttal.pdf, where the label distribution of Clipart domain differs substantially from other domains. The corresponding Jensen–Shannon divergence (Figure 5) verifies this and indicates a much more severe label shift for Clipart. For this sub-task, we compared our method against iMSDA, one of the strongest baselines in our experiments. Our method achieves an accuracy of $57.7$, outperforming iMSDA that achieves $57.1$, demonstrating the effectiveness of our approach under severe label shift.
>
> We will incorporate this discussion and additional experiment into the revision. Hope this addresses your concern.
>
> **Q2:** "Including the latest baselines would enhance the rigor of the analysis"
>
> **A2:** In light of your suggestion, we will include 5 additional recent baselines (after 2022) in the revision, i.e., CASR [4], TFFN [5], SSD [6], GeNRT [7], and iLCC-LCS [8]. The results are provided in the tables below, which indicate that our method achieves superior performance.
>
> - Office-Home dataset:
>
> | Method      | Ar   | Cl   | Pr   | Rw   | Avg  |
> |-------------|------|------|------|------|------|
> | CASR [4]    | 72.2 | 61.1 | 82.8 | 82.8 | 74.7 |
> | TFFN [5]    | 72.2 | 62.9 | 81.7 | 83.5 | 75.1 |
> | SSD [6]     | 72.5 | **64.5** | 81.2 | 83.2 | 75.4 |
> | **GAMA (Ours)** | **76.6** | 62.6 | **84.9** | **84.9** | **77.3** |
>
> - PACS dataset:
>
> | Method      | P    | A    | C    | S   | Avg  |
> |-------------|------|------|------|------|------|
> | GeNRT [7]    | 98.5 | 93.6 | 91.4 | 85.7 | 92.3 |
> | iLCC-LCS [8]    | 95.9 | 86.4 | 81.1 | 86.0 | 87.4 |
> | **GAMA (Ours)** | **98.8** | **93.7** | **92.8** | **89.3** | **93.7** |
>
> **Q3:** "Why did the authors choose a multi-VAE structure for implementation? Could the author provide more insights into their intuition behind this choice?"
>
> **A3:** Thanks for your question, which helps clarify the motivations behind our implementation. First, VAE provides a convenient way to model the distribution of latent variables. Second, compared to other generative models, VAEs make it easier to incorporate prior structural information (e.g., parent-child relationships) into our method. Using multiple VAEs further facilitates the integration of such structural priors. We will clarify this in the revision.
>
> ---
> **References:**
>
> [1] Le et al., On Label Shift in Domain Adaptation via Wasserstein Distance. arXiv, 2022.
>
> [2] Jang et al., Distribution Shift-Aware Prediction Refinement for Test-Time Adaptation. arXiv, 2024
>
> [3] Liu et al., Domain Generalization under Conditional and Label Shifts via Variational Bayesian Inference. In IJCAI, 2021.
>
> [4] Wang et al., Classaware sample reweighting optimal transport for multi-source domain adaptation. Neurocomputing, 2023.
>
> [5] Li et al., Transferable feature filtration network for multi-source domain adaptation. Knowledge-Based Systems, 2023.
>
> [6] Li et al., Multidomain adaptation with sample and source distillation. IEEE Transactions on Cybernetics, 2023.
>
> [7] Deng et al., Generative model based noise robust training for unsupervised domain adaptation. arXiv, 2023.
>
> [8] Liu et al., Identifiable latent causal content for domain adaptation under latent covariate shift. arXiv, 2024.

---

### Official Review · Reviewer_eX2F · 2025-03-16

**Overall Recommendation:** 4

**Summary:**

The manuscript presents a general representation-based approach for multi-source domain adaptation (GAMA). It aims to improve knowledge transfer across domains by leveraging theoretical identifiability results for latent variables and adapting a variational autoencoder (VAE)-based framework. The authors propose partitioning the Markov blanket into its parents, children, and spouses to enhance adaptation performance and validate their approach through theoretical guarantees and empirical evaluations on two benchmark datasets.

## update after rebuttal
The authors' answers clarified my doubts, so I confirm the positive assessment of the manuscript.

**Claims And Evidence:**

The manuscript makes strong theoretical claims regarding identifiability and its impact on domain adaptation. The theoretical framework is supported by a series of assumptions and theorems, including results on subspace identifiability of the Markov blanket and its components​. Empirical results on benchmark datasets substantiate the proposed method, demonstrating its effectiveness compared to existing approaches.

**Essential References Not Discussed:**

As far as I know, the authors have included relevant and recent references related to the subject matter.

**Experimental Designs Or Analyses:**

The experiments are well-structured, covering multiple datasets and baseline comparisons. However, additional baselines, particularly recent causal representation learning approaches, could further validate the method's effectiveness​.

**Methods And Evaluation Criteria:**

The authors employ a combination of variational autoencoders and deep neural networks to extract and learn latent representations that are invariant to domain shifts. Evaluation is conducted on standard datasets such as PACS and Office-Home​, using accuracy as the primary metric. The paper also includes an ablation study to analyze the impact of different components of the model​.

**Other Comments Or Suggestions:**

- Provide additional baselines from causal representation learning.
- Discuss the computational complexity of the approach.
- Clarify the generalizability of the method beyond the considered datasets.
- Page 7, line 372 first column. I think the authors would refer to Figure 2 instead of Figure 1.

**Other Strengths And Weaknesses:**

Strengths
- Strong theoretical foundation for identifiability.
- Clear motivation for partitioning the Markov blanket.
- Well-structured experimental validation.

Weaknesses
- Some theoretical assumptions may be restrictive.
- Lack of discussion on computational efficiency.
- No discussion on potential failure cases or limitations.

**Questions For Authors:**

1. How does the approach handle highly imbalanced domain shifts?
2. Would the method still be effective with fewer source domains?
3. Are there any practical limitations regarding computational resources for training?
4. Could the theoretical results be extended to other types of unsupervised domain adaptation problems?

**Relation To Broader Scientific Literature:**

The paper contextualizes its contributions well within domain adaptation and causal representation learning. It cites relevant works on domain adaptation, causal disentanglement, and latent variable identifiability​.

**Theoretical Claims:**

The manuscript provides rigorous theoretical backing, introducing multiple theorems to establish the identifiability of the joint distribution in the target domain. However, some assumptions, such as linear independence of certain distributions, may be restrictive and require further discussion​.

---

> ### Author Rebuttal · Authors · 2025-04-01
>
> We greatly appreciate the reviewer's time and valuable comments, many of which will help improve the clarity of our paper. Our responses to these comments are given below.
>
> **Q1:** "some assumptions, such as linear independence of certain distributions, may be restrictive and require further discussion"
>
> **A1:** Thanks for this comment. Intuitively, the sufficient change assumption requires the existence of multiple environments where the causal mechanisms among latent variables change sufficiently. These distributional changes, combined with the invariant mixing function, provide information for recovering the latent variables and their causal relationships. Note that this type of sufficient change assumption has been commonly adopted in the literature of nonlinear ICA and causal representation learning to establish identifiability under various settings. We will include this discussion in the revision.
>
> **Q2:** "Lack of discussion on computational efficiency" and "Are there any practical limitations regarding computational resources for training?"
>
> **A2:** We train our model using a NVIDIA A100-SXM4-40GB GPU. For Office-Home dataset, the batch size is set to 32, and the model is trained for 70 epochs, which takes approximately 160 minutes. The peak memory usage is around 35 GB. The majority of the computational cost comes from the ResNet-50 backbone, as we only add several lightweight MLP layers after it. We will discuss this and report the running time for all datasets in the revision.
>
> **Q3:** "No discussion on potential failure cases or limitations"
>
> **A3:** A limitation, as discussed in Q2, is that our method may require a relatively long training time. However, we believe this trade-off is justified given the performance improvements it achieves. We will discuss this in the revision.
>
> **Q4:** "Provide additional baselines from causal representation learning"
>
> **A4:** There are only a few works from causal representation learning that address domain adaptation. The only work we are aware of is iMSDA (Kong et al., 2022), which we have included as a baseline. Please kindly let us know if there is other suitable causal representation learning baseline you think we could compare to.
>
> **Q5:** "Clarify the generalizability of the method beyond the considered datasets"
>
> **A5:** Our method is designed for general domain adaptation, effectively handling various types of distribution shifts by learning latent representations that are most relevant for adaptation and capturing distribution shifts through low-dimensional representations. The datasets we used—Office-Home and PACS—are widely adopted benchmarks that span diverse domains and distribution shifts, providing strong empirical support for its generalizability.
>
> To further evaluate the generalizability, we have conducted additional experiments under more severe label shifts; see our response to Q1 for Reviewer d2cE. It is observed that our method continues to perform well, demonstrating its effectiveness even in more challenging settings.
>
> **Q6:** "Page 7, line 372 first column. I think the authors would refer to Figure 2 instead of Figure 1"
>
> **A6:** We will fix the typo in the revision.
>
> **Q7:** "How does the approach handle highly imbalanced domain shifts?"
>
> **A7:** Based on your question, we interpret "highly imbalanced domain shifts" as referring to large distribution shifts between domains. Under this interpretation, our approach remains applicable as long as the assumptions hold. Specifically, the more imbalanced the domain shifts are, the more domains may be required to satisfy our theoretical assumptions. We will incorporate this discussion in the revised manuscript. If you intended a different interpretation of "imbalanced domain shifts," we would greatly appreciate if you could kindly let us know.
>
> **Q8:** "Would the method still be effective with fewer source domains?"
>
> **A8:** While our theoretical result assumes multiple domains, our method remains highly effective even with only a limited number of source domains. This is demonstrated by the strong performance in our experiments, where only a limited number (i.e., three) of source domains are available. We will include this discussion in the revision.
>
> **Q9:** "Could the theoretical results be extended to other types of unsupervised domain adaptation problems?"
>
> **A9:** Our theoretical results are already quite general, as they accommodate (1) various types of distribution shifts, where changes may occur anywhere in the latent space, and (2) arbitrary relations among latent variables. To address your question, while our results assume a single target domain, **it can be naturally extended to multi-target domain adaptation** by learning distinct $P_{new}$ for each target domain. This extension is feasible because our framework learns compact latent representations that capture distribution shifts relative to the prediction task. We will include a discussion of this extension in Section 4.3.

---

> > ### Comment · Reviewer_eX2F · 2025-04-02
> >
> > I thank the authors for their responses that clarified my doubts, so I confirm the positive judgment on the manuscript.

---

> > > ### Author Response · Authors · 2025-04-07
> > >
> > > Thanks for your recognition and constructive feedback. We will incorporate them into the manuscript. Please feel free to let us know if you have further questions. Thank you!

---

### Official Review · Reviewer_eV3Q · 2025-03-17

**Overall Recommendation:** 3

**Summary:**

The paper proposes a multi-source domain adaptaion approach. It is a generative based approach which projects feature representation into a latent space using VAE. The latent space (Markov blanket) is then partitioned into the subspace of label’s parents, children and spouses. Next, two VAE are used to Z_pa and Z_sps to learn \theta_Y and \theta_ch. A classification loss is added to train on input labels.

**Claims And Evidence:**

Yes

**Essential References Not Discussed:**

Yes

**Experimental Designs Or Analyses:**

Yes

**Methods And Evaluation Criteria:**

Yes

**Other Comments Or Suggestions:**

NA

**Other Strengths And Weaknesses:**

Strengths:
1. The method is novel and has a detailed theoretical background and explanation.
2. Approach achieves good results on Office Home and PACS datasets.


Weakness:
1. More experiments would be better. Proposed approach is evaluated on two datasets (std should be added to the results). It is suggested to include at least 3 datasets for reliability. Domainnet is a popular dataset for multi-source domain adaptation.
2. Limited analysis. Analysis performed on the approach is not enough. Some experiments related to the latent space of features and VAE would be nice to visualize.
3. The approach has too many hyper-parameters. How and what these values are set to is not available in the paper.

**Questions For Authors:**

See weakness.

**Relation To Broader Scientific Literature:**

The paper presents an incremental solution to multi-source domain adaptation problem.

**Theoretical Claims:**

Glanced over them.

---

> ### Author Rebuttal · Authors · 2025-04-01
>
> We sincerely thank the reviewer for the time dedicated to reviewing our paper and the valuable comments. We have tried to address all the concerns in the following.
>
> **Q1:** "More experiments would be better" and "Domainnet is a popular dataset for multi-source domain adaptation"
>
> **A1:** Thank you for your insightful suggestion. Following your suggestion, we are currently conducting experiments for the DomainNet dataset. The dataset contains 6 different sub-tasks. Due to time constraints of the rebuttal period, some of these experiments are still ongoing, and we have finished the experiments for the Clipart sub-task. The results are provided in the table below, which indicate that our method outperforms all other baselines. Note that to save space for author's response, the references of the baselines, if not specified, can be found in our paper (e,g., Table 1). We will include the complete results for different sub-tasks in the revised manuscript. Hope this addresses your concern.
>
> | Method      | Clipart |
> |-------------|------|
> | Source Only    | 52.1 |
> | DANN    | 60.6 |
> | DCTN    | 48.6 |
> | MCD    | 54.3 |
> | M3SDA    | 58.6 |
> | CMSS    | 64.2 |
> | LtC-MSDA    | 63.1 |
> | ADDA [1]    | 47.5 |
> | ML_MSDA [2]    | 61.4 |
> | meta-MCD [3]    | 62.8 |
> | PFSA [4]    | 64.5 |
> | SSD [5]    | 67.2 |
> | **GAMA (Ours)** | **69.2** |
>
> **Q2:** "std should be added to the results"
>
> **A2:** Thanks for this suggestion. We will include the standard deviation for all datasets in the revised manuscript. Due to space constraints of author's response, let us provide the results with standard deviation for Office-Home dataset in the table below.
>
> | Method | Ar | Cl | Pr | Rw | Avg |
> |-------------|------|------|------|------|------|
> | DAN    | 68.3±0.5 | 57.9±0.7 | 78.5±0.1 | 81.9±0.4 | 71.6 |
> | Source Only    | 64.6±0.7 | 52.3±0.6 | 77.6±0.2 | 80.7±0.8 | 68.8 |
> | DANN    | 64.3±0.6 | 58.0±1.6 | 76.4±0.5 | 78.8±0.5 | 69.4 |
> | DCTN    | 66.9±0.6 | 61.8±0.5 | 79.2±0.6 | 77.8±0.6 | 71.4 |
> | MCD    | 67.8±0.4 | 59.9±0.6 | 79.2±0.6 | 80.9±0.2 | 72.0 |
> | DANN+BSP | 66.1±0.3 | 61.0±0.4 | 78.1±0.3 | 79.9±0.1 | 71.3 |
> | M3SDA    | 66.2±0.5 | 58.6±0.6 | 79.5±0.5 | 81.4±0.2 | 71.4 |
> | iMSDA    | 75.4 ± 0.9 | 61.4 ± 0.7 | 83.5 ± 0.2 | 84.5 ± 0.4 | 76.2 |
> | **GAMA (Ours)** | **76.6±0.1** | **62.6±0.6** | **84.9±0.1** | **84.9±0.1** | **77.3** |
>
> **Q3:** "Limited analysis" and "Some experiments related to the latent space of features and VAE would be nice to visualize"
>
> **A3:** We appreciate your helpful suggestion, which will aid our understanding of the method. In light of your suggestion, we have conducted visualizations of the latent space of features and VAE. Specifically, the t-SNE visualizations of the learned features on the Clipart task from the Office-Home dataset are available in Figure 6 in the anonymized repository: https://anonymous.4open.science/r/icml-rebuttal-gama/rebuttal.pdf, which demonstrate the effectiveness of our method at aligning the source and target domains while preserving discriminative structures. We will include the visualizations in the revised manuscript, and hope this addresses your concern.
>
> **Q4:** "The approach has too many hyper-parameters. How and what these values are set to is not available in the paper."
>
> **A4:** Thanks for your comment. We followed existing work [6] and selected hyperparameters that lead to optimal performance for each task. Due to space constraints of author's response, the values of these hyperparameters are provided in Table 1 of the anonymized repository: https://anonymous.4open.science/r/icml-rebuttal-gama/rebuttal.pdf. We will include these details in the revised manuscript for clarity.
>
> **We want to thank the reviewer again for all the valuable feedback.**
>
> ---
> **References:**
>
> [1] Tzeng et al., Adversarial discriminative domain adaptation. In CVPR, 2017.
>
> [2] Li et al., Mutual learning network for multi-source domain adaptation. arXiv preprint arXiv:2003.12944, 2020.
>
> [3] Li et al., Online meta-learning for multi-source and semi-supervised domain adaptation. In ECCV, 2020.
>
> [4] Fu et al., Partial feature selection and alignment for multi-source domain adaptation. In CVPR, 2021.
>
> [5] Li et al., Multidomain adaptation with sample and source distillation. IEEE Transactions on Cybernetics, 2023.
>
> [6] Kong et al., Partial disentanglement for domain adaptation. In ICML, 2022.

---

### Decision · Program_Chairs · 2025-05-01

**Decision:**

Accept (poster)

**Comment:**

This paper presents a novel contribution for multi-source Domain Adaptation based on a identifiability framework and the use of VAE.


During evaluation, the novelty of the paper and the strong theoretical background  of the contribution have been noted, however some limitations in the experiments, the analysis and the discussion were also raised.
During rebuttal, authors have provided different answers to the reviewers.
After the rebuttal, the three reviewers maintained a positive evaluation (2 weak accepts and 1 accept) and agreed on the recommendation to accept the paper.